# mGluR5 is transiently confined in perisynaptic nanodomains to shape synaptic function

Nicky Scheefhals [1], Manon Westra [1] & Harold D. MacGillavry [1]✉

The unique perisynaptic distribution of postsynaptic metabotropic glutamate receptors (mGluRs) at excitatory synapses is predicted to directly shape synaptic function, but mechanistic insight into how this distribution is regulated and impacts synaptic signaling is lacking. We used live-cell and super-resolution imaging approaches, and developed molecular tools to resolve and acutely manipulate the dynamic nanoscale distribution of mGluR5. Here we show that mGluR5 is dynamically organized in perisynaptic nanodomains that localize close to, but not in the synapse. The C-terminal domain of mGluR5 critically controlled perisynaptic confinement and prevented synaptic entry. We developed an inducible interaction system to overcome synaptic exclusion of mGluR5 and investigate the impact on synaptic function. We found that mGluR5 recruitment to the synapse acutely increased synaptic calcium responses. Altogether, we propose that transient confinement of mGluR5 in perisynaptic nanodomains allows flexible modulation of synaptic function.

Precise modulation of glutamatergic synaptic transmission is critical for the execution of cognitive processes. Glutamatergic transmission is mediated by two types of postsynaptic glutamate receptors: the ionotropic glutamate receptors (iGluRs), including the AMPA and NMDA-type receptors, and the group I metabotropic glutamate receptors (mGluRs), mGluR1 and mGluR5. While iGluRs carry the majority of fast signal transmission across synapses, mGluRs modulate the efficacy of synaptic signaling on longer time scales by coupling to a variety of effector systems that collectively modulate synaptic transmission and plasticity[1,2]. Postsynaptic group I mGluRs canonically signal through IP3-mediated calcium release from internal stores, but also through modulation of NMDA receptors[3]. The contribution of mGluRs to glutamatergic signaling has been found to be critical for cognitive functions such as attention and learning and memory, and disrupted mGluR signaling has been implicated in diverse neurological disorders[4]. Yet, the precise organization of mGluRs within the perisynaptic zone and the underlying mechanisms, critical to efficiently modulate synaptic transmission, are still poorly understood.

Key to the modulation of receptor activation is their subsynaptic organization and alignment with presynaptic vesicle release sites. iGluRs organize in nanodomains within the postsynaptic density (PSD), aligned with vesicle release sites within the presynaptic active zone, increasing the strength of a synaptic response[5–10]. In contrast, group I mGluRs are enriched in the perisynaptic zone, an annular ring of ~200 nm surrounding the PSD, considerably further away from vesicle release events[11–13]. A single vesicle release event induces a very local and transient glutamate gradient only activating the opposing iGluRs due to the low affinity of AMPARs and mGluRs for glutamate (0.5–2 mM) and the slow glutamate binding rate of NMDARs[14–16]. Kinetic profiling of mGluR activation predicts that high frequency or repetitive stimulation is required for glutamate to reach sufficient concentrations in the synaptic cleft to also activate the perisynaptic mGluRs[17–19]. Hence, the spatial segregation of these functionally distinct receptor types allows for the precise temporal control of synaptic transmission and plasticity. The synaptic density and organization of receptors are not static, but highly dynamic, governed by processes that affect receptor mobility, such as lateral diffusion, endocytosis and exocytosis, and immobilization to synaptic structures[20,21]. Disrupted mGluR mobility has been implicated in neurological and neurodegenerative disorders[22–24]. Thus, an understanding of the dynamic

[1]Cell Biology, Neurobiology and Biophysics, Department of Biology, Faculty of Science, Utrecht University, 3584 CH Utrecht, The Netherlands.
✉e-mail: h.d.macgillavry@uu.nl

organization of mGluRs is critical to provide new insights into the mechanisms underlying synaptic transmission in both physiological and pathophysiological conditions.

Here, we use complementary super-resolution imaging approaches and show that mGluR5 is largely excluded from the core of the PSD and is preferentially confined in perisynaptic nanodomains. We demonstrate that the C-terminal domain of mGluR5 mediates perisynaptic confinement, but also prevents synaptic entry of mGluR5, even when forced to interact with synaptic scaffolds. We furthermore show that acute disruption of the perisynaptic organization of mGluR5 deregulates calcium signaling in spines.

## Results

### mGluR5 is enriched in spines but largely excluded from the synapse

To study the distribution of surface-expressed mGluR5 in neurons, we transfected hippocampal neurons with mGluR5 coupled to an extracellular super-ecliptic pHluorin (SEP) tag, additionally labeled with a cell-impermeable GFP nanobody conjugated to Atto647N. The expression of mGluR5 was observed throughout neurons, but was most prominent in the dendritic shaft and spines (Fig. 1a). mGluR5 was significantly enriched in spines compared to a mCherry fill (mGluR5: $1.49 \pm 0.035$ and mCherry: $0.74 \pm 0.04$, Fig. 1b, c), but significantly less enriched compared to the PSD scaffolding protein Homer1c (mGluR5: $1.49 \pm 0.029$ and Homer1c: $2.16 \pm 0.066$; Fig. 1c, d). Homer1c overexpression to mark the PSD did not affect mGluR5 enrichment in spines (Fig. 1c, light gray bars). Next, we used gated stimulated emission depletion (gSTED) microscopy to assess mGluR5 localization relative to the PSD. Foremost, we found that mGluR5 is largely excluded from the PSD, confirming early EM studies[11–13] (Fig. 1e, f). Also, two-color gSTED imaging of mGluR5 and the PSD, labeled with a PSD-95 antibody, revealed minimal co-localization between mGluR5 and the PSD (Fig. S1a–d).

We always selected neurons with moderate overexpression levels of mGluR5 (Fig. S1e), and found that the median mGluR5 overexpression was ~2 times higher compared to endogenous mGluR5 levels (Fig. S1e–g). Endogenous mGluR5 labeled with an mGluR5 antibody was similarly excluded from PSD-95 immunolabelled synapses compared to overexpressed mGluR5 (Fig. S1h–k), and rather localized close to phalloidin staining F-actin, known to be enriched in the perisynaptic zone[25] (Fig. S1l–n). Furthermore, we endogenously tagged mGluR5 with an extracellular GFP-tag using the ORANGE CRISPR/Cas9-based knock-in toolbox[9] (Fig. 1i). gSTED imaging of the surface-labeled mGluR5 knock-in also revealed mGluR5 distribution throughout the dendrite, with preferential perisynaptic localization in spines (Fig. 1j–m). Notably, the localization observed for mGluR5 is markedly different from other glutamate receptors, including AMPA receptors. The AMPA receptor subunit GluA2 co-localized with the PSD marked by Homer1c (Fig. S1o–r) and localized in subsynaptic domains spatially segregated from mGluR5 shown with two-color gSTED microscopy (Fig. S1s–v).

### mGluR5 is organized in perisynaptic nanodomains

To resolve the nanoscale perisynaptic distribution of mGluR5 we used two-color single-molecule localization microscopy (SMLM) on neurons transfected with SEP-mGluR5, labeled with an anti-GFP nanobody coupled to Alexa647, and mEos3.2-tagged PSD$_{FingR}$ to label the PSD[26] (Figs. 2a, b and S2a). PSDs were identified using density-based spatial clustering of applications with noise (DBScan)[27] on the PSD$_{FingR}$-mEos3.2 localizations (Fig. 2c). Consistent with our previous observations, we found that most mGluR5 localizations are within 200 nm from the PSD border (Fig. 2d). To investigate this more closely, we mapped the localizations of mGluR5 and PSD$_{FingR}$ in eight incremental rings proportionally scaled to the PSD border to normalize for PSD size (Figs. 2e and S2b)[28]. As expected, almost all PSD$_{FingR}$ molecules were

found within the two inner synaptic rings and were almost absent from the surrounding rings. In contrast, we found that mGluR5 localizations were enriched in the three perisynaptic rings, compared to the synaptic and extrasynaptic rings (Fig. 2f, also see Fig. S2c for absolute number of localizations).

In these two-color SMLM experiments we observed that mGluR5 was not homogeneously distributed in the perisynaptic region. Indeed, using DBScan we found that mGluR5 is concentrated in subsynaptic nanodomains. These nanodomains were most frequently found within the perisynaptic region, with a median border-to-centroid distance from PSD to mGluR5 nanodomains of 240 nm (Fig. 2g, h). The median area of individual mGluR5 nanodomains was $6.0 \times 10^3$ nm² (95% CI [5.2 7.2] $\times 10^3$; Fig. 2i) and 115 nm in length and 83 nm in width (length: 95% CI [111 120] and width: (95% CI [81 88], full width tenth maximum (FWTM); Fig. 2j). The total mGluR5 nanodomain area per PSD slightly correlated with PSD area (Fig. S2d). Using SR-Tesseler, another quantitative approach based on Voronoi diagrams to segment and quantify protein organization (Fig. S2e, f)[29], we confirmed the perisynaptic mGluR5 nanodomains observed using DBScan (Fig. S2g). The SR-Tesseler approach detected more and significantly smaller clusters compared to DBScan (median clusters area: DBScan: $6.3 \times 10^3$ nm² and SR-Tesseler: $2.1 \times 10^3$ nm²; Fig. S2h), however, this is inherent to the method used to outline the clusters and the different input parameters (see M&M for details). Even though we used stringent criteria to ensure that nanodomains consisted of a considerable amount of receptors, we set out to exclude the possibility that the nanodomains represent dimeric receptors. We used SR-Tesseler without set cluster criteria to detect nano-objects to discriminate between dimeric and clustered receptors in the dendritic shaft and spines. In both the dendritic shaft and spines we found many objects otherwise excluded from the analysis, with the smallest objects likely representing mGluR dimers. In addition, the objects detected in spines were distinct from those on the dendrite as the spine objects were significantly larger in area (median object area: spines: $0.39 \times 10^3$ nm² and dendrites: $0.18 \times 10^3$ nm²; Fig. S2i, j). Thus, our two-color SMLM experiments revealed a high degree of organization of mGluR5, demonstrating that mGluR5 is enriched in distinct nanodomains that preferentially localize in the perisynaptic zone.

### The spatial distribution of mGluR5 diffusion at and around the synapse is highly heterogeneous

We observed a remarkable heterogeneous perisynaptic distribution of mGluR5, however, we have little insight into whether mGluR5 is stably anchored at perisynaptic sites or only transiently visits these perisynaptic nanodomains. Nevertheless, such information is critical to better understand how mGluR5 contributes to synaptic signaling. To address this, we used a single-molecule tracking (SMT) approach called universal point accumulation in nanoscale topography (uPAINT)[30] to study the subsynaptic mobility of mGluR5. Neurons were co-transfected with SEP-mGluR5 and Homer1c-mCherry to mark the PSD (Fig. 3a). SMT was performed using an anti-GFP nanobody coupled to Atto647N that stochastically labeled individual SEP-tagged receptors, providing a map of mGluR5 mobility (Fig. 3b). The diffusion coefficients of mGluR5 trajectories in spines were significantly lower than in dendrites (median $D_{eff}$ spines: 0.022 μm²/s, dendrites: 0.036 μm²/s; Fig. S3c, d). To further differentiate between mGluR5 diffusion at different synaptic subregions, we used the Homer1c-mCherry channel to mark the synaptic region (PSD mask), as well as an annulus surrounding the PSD by expanding the PSD mask with 200 nm to mark the perisynaptic zone. Importantly, we found that mGluR5 diffusion was similar in neurons expressing mCherry and Homer1c, indicating that Homer1c overexpression does not alter mGluR5 diffusion (Fig. S3a–f). Trajectories were categorized as synaptic, perisynaptic, or transient perisynaptic, all associating with the synapse and/or perisynaptic zone but to different extents (see M&M for details)

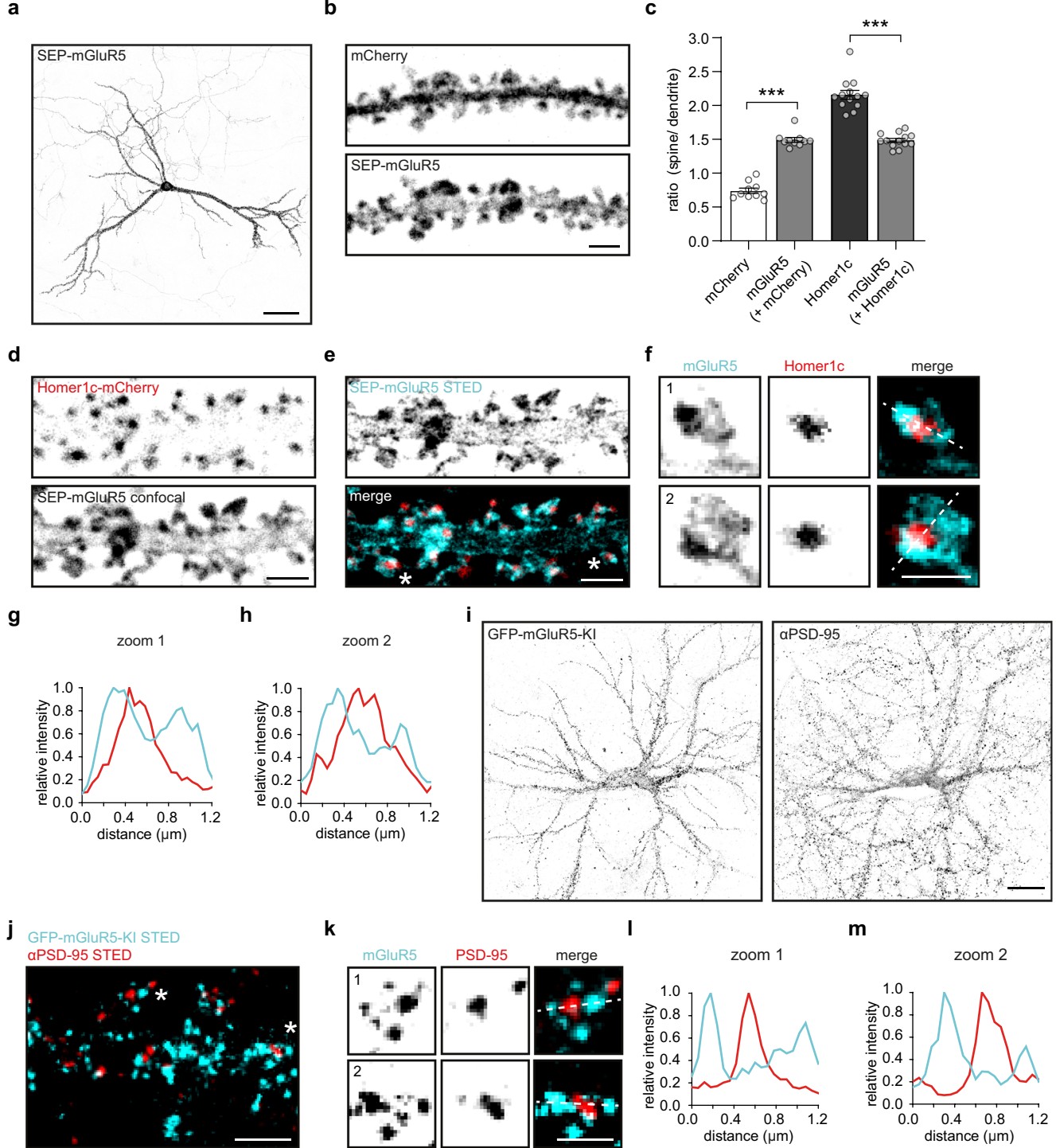

**Fig. 1 | mGluR5 is enriched in spines but largely excluded from synapses in hippocampal neurons. a** Hippocampal neuron expressing SEP-mGluR5. Scale bar, 50 μm. **b** Representative confocal image of dendrite expressing mCherry and SEP-mGluR5, surface-labeled with an anti-GFP nanobody Atto647N. Scale bar, 2 μm. **c** Quantification of the ratio of spine over dendrite intensity of mCherry, surface SEP-mGluR5 co-expressing mCherry ($n = 10$, $p < 0.0001$; two-sided paired $t$ test), Homer1c-mCherry and surface SEP-mGluR5 co-expressing Homer1c-mCherry ($n = 13$, $p < 0.0001$; two-sided paired $t$ test). **d** Representative confocal image of dendrite expressing Homer1c-mCherry and SEP-mGluR5, surface-labeled with an anti-GFP nanobody Atto647N. Scale bar, 2 μm. **e** gSTED imaging of SEP-mGluR5 surface-labeled with an anti-GFP nanobody Atto647N (cyan) and the merged image showing the relative localization to confocal-resolved Homer1c-mCherry (red), shown in *d*. Scale bar, 2 μm. This experiment was replicated in cultures from more than three independent preparations of hippocampal neurons. **f** Zooms of dendritic spines indicated in **e** with asterisks. Scale bar, 1 μm. **g** Line profiles of spine 1 and **h** spine 2, indicated with dotted line in **f. i** Hippocampal neuron with an ORANGE GFP knock-in (KI) endogenously tagging mGluR5 at the N-terminus, enhanced with anti-GFP Alexa488 labeling (left), co-stained for anti-PSD-95 Alexa594 (right). Scale bar, 20 μm. **j** Representative two-color gSTED image of dendrite with GFP-mGluR5 KI stained with anti-GFP Alexa488 to label surface-expressed receptors (cyan) and anti-PSD-95 Alexa594 (red). Scale bar, 2 μm. This experiment was replicated in cultures from three independent preparations of hippocampal neurons. **k** Zooms of dendritic spines indicated in **j** with asterisks. Scale bar, 1 μm. **l** Line profiles of spine 1 and **m** spine 2, indicated with dotted line in **k**. Data are represented as means ± SEM. ***$p < 0.001$. Source data are provided as a Source Data file.

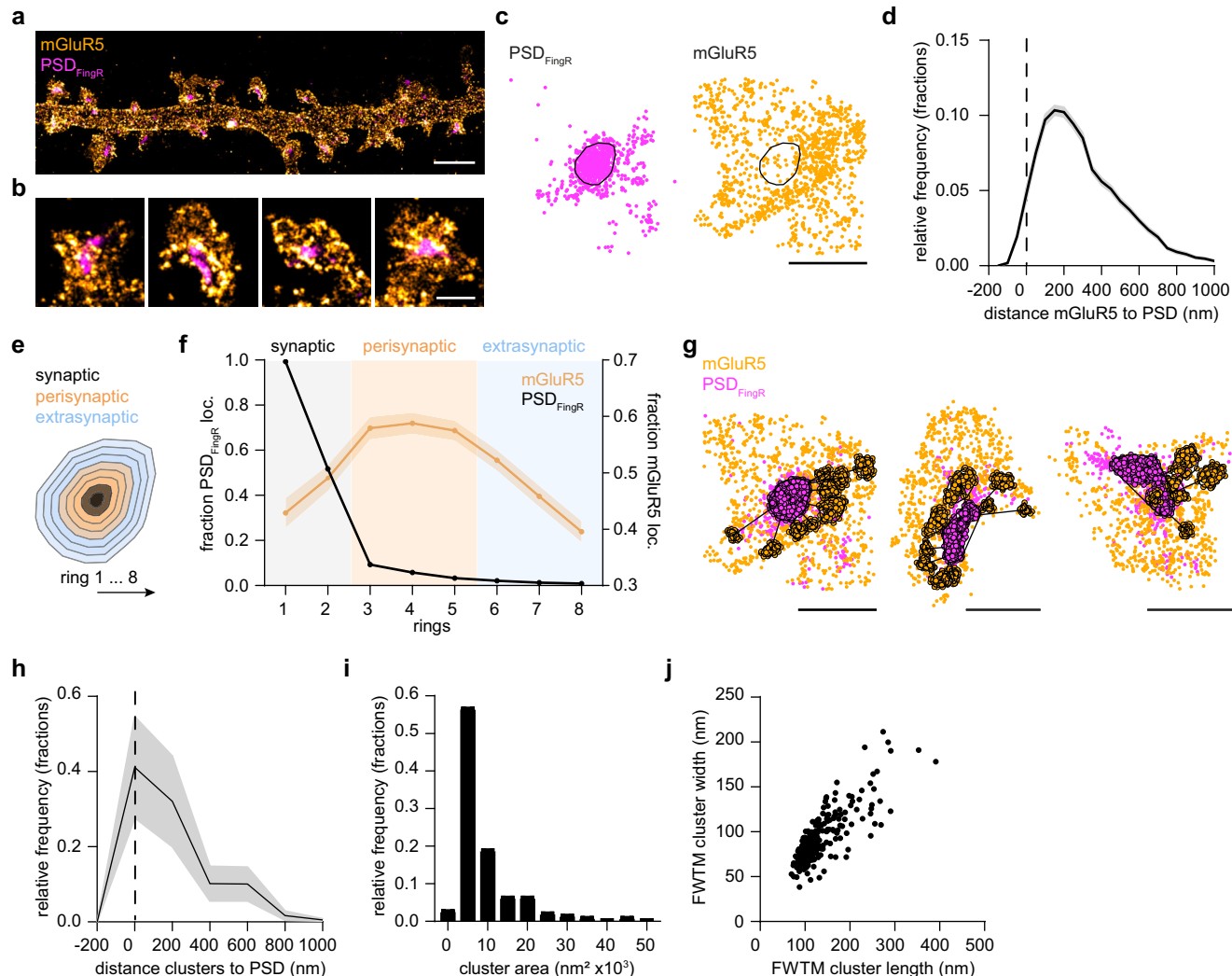

**Fig. 2 | mGluR5 is organized in perisynaptic nanodomains. a** Reconstruction of single-molecule localizations obtained for SEP-mGluR5 anti-GFP nanobody Alexa647 using dSTORM (orange hot) and PSD$_{FingR}$-mEos3.2 using PALM (cyan hot). Same dendritic region is shown in Fig. S2a. Scale bar, 2 μm. **b** Zooms of spines shown in **a**. Scale bar, 500 nm. **c** Representative spine with single-molecule localizations of PSD$_{FingR}$ (cyan) and mGluR5 (orange) with indicated PSD border (black line) determined using DBScan. Scale bar, 500 nm. **d** Relative frequency distribution (fractions) of the distance of individual mGluR5 localizations to the PSD border (n = 13 neurons, 253 PSDs). **e** For each PSD, eight rings, proportionally scaled based on its PSD border, defined the synapse (ring 1 and 2; black), perisynaptic zone (ring 3–5; orange), and extrasynaptic region (ring 6–8; blue). **f** Fraction of PSD$_{FingR}$ (black; plotted on left y axis) and mGluR5 (orange; plotted on right y-axis) localizations in rings 1–8. For each PSD, the number of localizations was normalized to the maximum number per ring, and the area per ring was calculated and corrected for. **g** Example spines with mGluR5 localizations (orange) belonging to clusters (black outline) as determined using DBScan, relative to PSD$_{FingR}$ localizations (magenta). Scale bars, 500 nm. **h** Relative frequency distribution (fractions) of the distance from the center of mGluR5 clusters to the border of the PSD (as indicated in **g** by the black lines) (n = 273 clusters). **i** Relative frequency distribution (fractions) of the mGluR5 cluster area. **j** FWTM analysis comparing the width and length (in nm) of individual mGluR5 clusters. Data are represented as means ± SEM. Source data are provided as a Source Data file.

(Fig. S3g). The transient perisynaptic trajectories are largely extrasynaptic and only shortly overlap with the perisynaptic zone to capture the extrasynaptic spine population. Since we were interested in mGluR5 dynamics within spines, trajectories without overlap with the synapse and/or perisynaptic zone were not included for further analysis. We found a large fraction of perisynaptic mGluR5 trajectories and a significantly smaller fraction of mGluR5 trajectories within the synapse (synaptic: 0.15 ± 0.01, perisynaptic: 0.57 ± 0.01, transient perisynaptic: 0.28 ± 0.01; Figs. 3c, s and S3h), consistent with the mGluR5 distribution found using SMLM and gSTED microscopy. We hypothesized that mechanisms underlying the perisynaptic mGluR5 nanodomains likely influence receptor diffusion. Indeed, the large pool of perisynaptic mGluR5 diffused much slower compared to the mGluR5 trajectories that only transiently associated with the perisynaptic zone (median D$_{eff}$: synaptic: 0.014 μm$^2$/s, perisynaptic:

0.023 μm$^2$/s, transient perisynaptic: 0.042 μm$^2$/s; Fig. 3e, f), suggesting that mGluR5 surface mobility is specifically regulated at perisynaptic sites. The small fraction of mGluR5 within the synapse diffused at even lower rates, suggesting that although a small fraction of mGluR5 enters the PSD, these receptors are severely hindered in their diffusion. Notably, most receptors that entered the perisynaptic zone remained there for the full duration of the observation time (here termed 'captured': 0.61 ± 0.01), or left the perisynaptic zone but returned to the perisynaptic zone ('returned': 0.29 ± 0.009). Only a very small fraction of perisynaptic tracks escaped the perisynaptic zone ('escaped': 0.11 ± 0.007; Fig. 3g). Thus, corroborating our SMLM data, these observations suggest that there is an underlying mechanism that hinders free diffusion of mGluR5 specifically in the perisynaptic zone, effectively containing mGluR5 within the perisynaptic zone. Indeed, the MSD plots indicate that perisynaptic mGluR5 receptors undergo

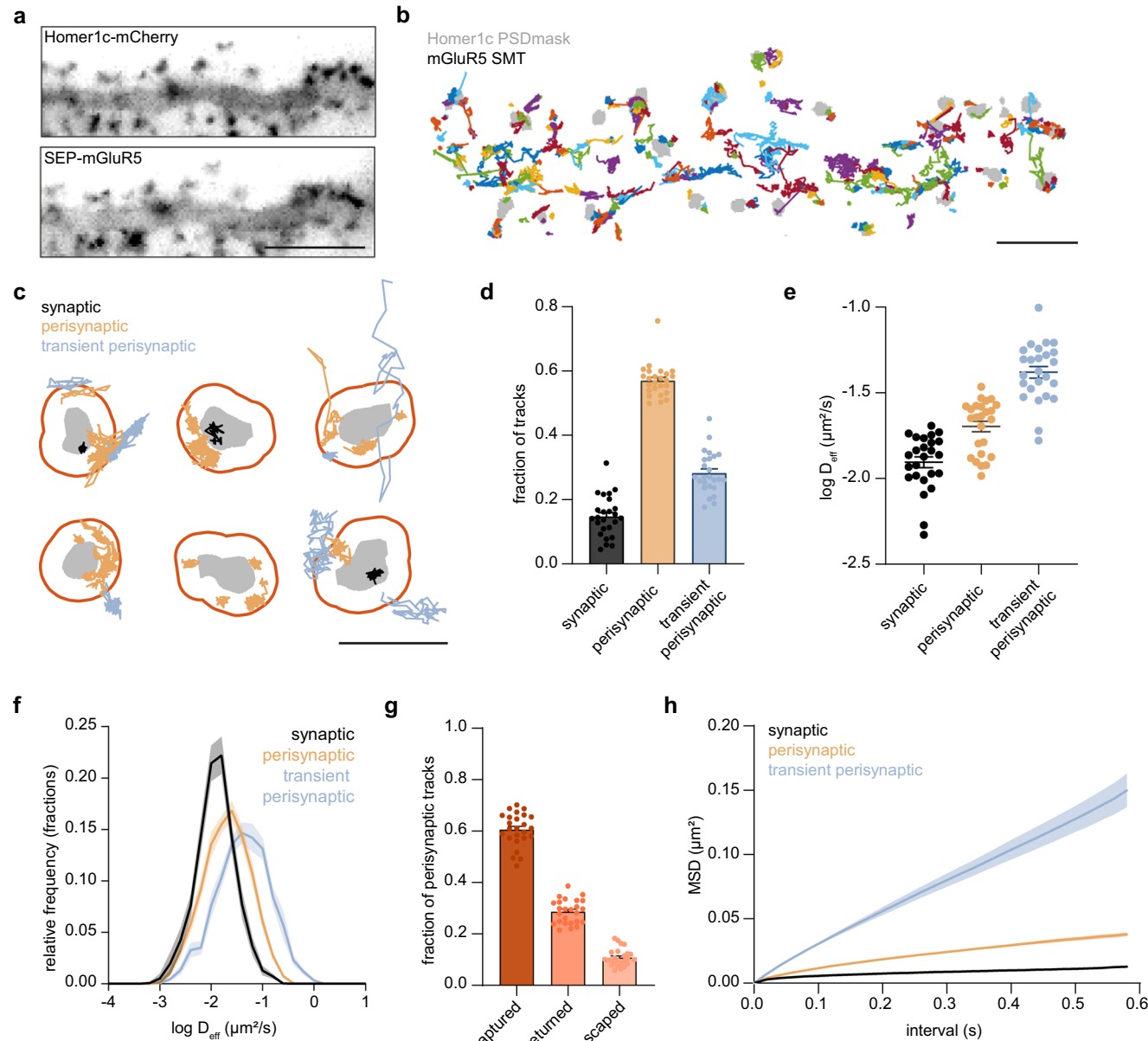

**Fig. 3 | Distribution of mGluR5 diffusion in spines is heterogeneous. a** Widefield of dendrite expressing Homer1c-mCherry and SEP-mGluR5. Scale bar, 5 μm. **b** Single-molecule trajectories (SMTs) of mGluR5 (each trajectory is assigned a random color) relative to the Homer1c PSD mask (gray) in the same dendrite as shown in a. Scale bar, 2 μm. **c** Example PSDs (gray) with their perisynaptic zone (orange ring) and mGluR5 SMTs color-coded for their subsynaptic localization. Scale bar, 1 μm. **d** Fraction of synaptic, perisynaptic, and transient perisynaptic mGluR5 SMTs ($n = 25$ neurons). **e** Mean log $D_{eff}$ per neuron of synaptic, perisynaptic, and transient perisynaptic mGluR5 SMTs ($n = 25$ neurons). **f** Relative frequency distributions of $D_{eff}$s of individual synaptic, perisynaptic, and transient perisynaptic mGluR5 SMTs. **g** Fraction of perisynaptic SMTs that stay within perisynaptic/synaptic region once entered (captured), exited at least once, but end up staying inside the perisynaptic/synaptic region (returned) and perisynaptic tracks that escaped the perisynaptic/synaptic region (escaped) ($n = 25$ neurons). **h** Mean MSD curve over time of synaptic, perisynaptic, and transient perisynaptic mGluR5 SMTs. Data are represented as means ± SEM. Source data are provided as a Source Data file.

anomalous diffusion, in contrast to the transient perisynaptic mGluR5 trajectories that seem to undergo Brownian diffusion (Fig. 3h).

**mGluR5 is transiently confined in perisynaptic nanodomains**

To further delineate how mGluR5 diffusion is locally controlled at perisynaptic sites, we next investigated the spatial distribution of mGluR5 immobilization and confinement. First, we classified mGluR5 trajectories as either mobile or immobile based on the ratio between the radius of gyration and the mean displacement per time step of individual trajectories[31]. We then mapped the immobile and mobile trajectories relative to the Homer1c PSD mask (Figs. 4a, b and S4a, b).

We found that the majority of mGluR5 trajectories was immobile (fraction of tracks: immobile: 0.65 and mobile: 0.35; Fig. S4c), with an expected diffusion coefficient slower than the mobile trajectories (median $D_{eff}$ immobile trajectories: 0.016 μm²/s and mobile trajectories: 0.050 μm²/s; Fig. S4d, e). Next, we sought to investigate whether the mobile mGluR5 trajectories undergo transient periods of confinement. We therefore estimated the confinement index $L$, which relates to the probability that a molecule undergoes confined diffusion in a region of radius $R$ for a period of time $t$[32–34]. This analysis revealed that a substantial fraction of the mobile mGluR5 trajectories (~40%) undergoes transient confinement with single trajectories displaying alternating

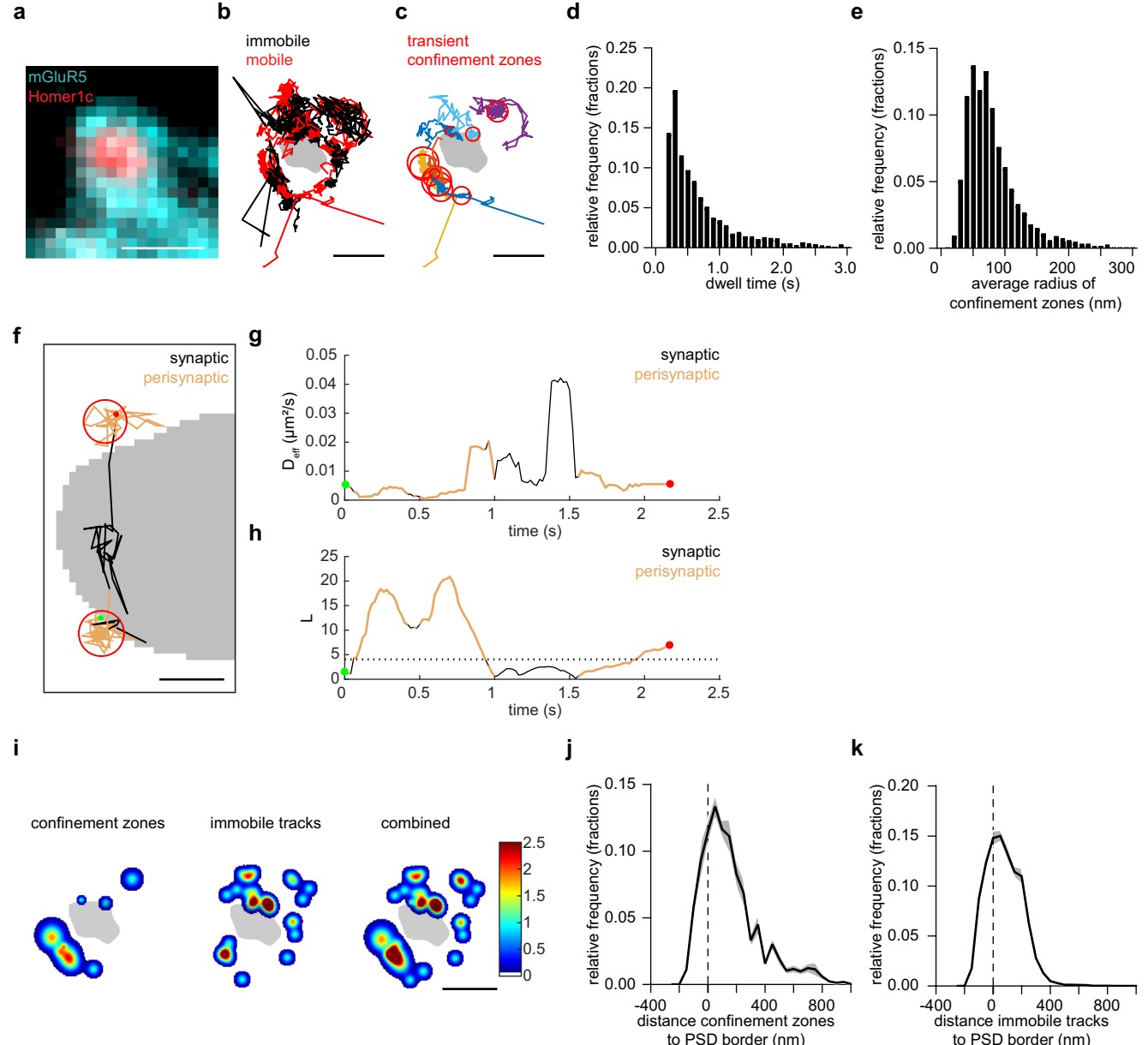

**Fig. 4 | mGluR5 is transiently confined, but also immobilized at the perisynaptic zone. a** Widefield image of a dendritic spine expressing SEP-mGluR5 (cyan) and Homer1c-mCherry (red). (larger ROI shown in Fig. S4a). Scale bar, 1 μm. **b** The same spine as in **a** showing immobile (black) and mobile (red) SMTs of mGluR5 relative to the Homer1c PSD mask (larger ROI shown in Fig. S4b). Scale bar, 500 nm. **c** Transient confinement zones (red circles) of the mobile trajectories (random colors) shown in B (larger ROI shown in Fig. S4f). Scale bar, 500 nm. **d** Relative frequency plots of the dwell time (s) and **e** average radius of confinement zones for mGluR5 SMTs. **f** Example trajectory (assigned to perisynaptic fraction) that undergoes transient confinement in the perisynaptic zone, color-coded for entering the synapse (black) and perisynaptic zone (orange) and the confinement zones (red). The trajectory starts (green dot) and ends (red dot) in perisynaptic transient confinement zones. Scale bar, 100 nm. **g** The diffusion coefficient and **h** confinement index $L$ over time for the trajectory shown in **f**, using the same color-coding. **i** The same example synapse as in **a**–**c** with hotspots of transient confinement zones, immobile tracks, and both images combined, color-coded for the frequency of confinement zones and/or immobile tracks (larger ROI shown in Fig. S4i). Scale bar, 500 nm. **j** Relative frequency distribution of the distance of confinement zones of mGluR5 and **k** center of immobile mGluR5 trajectories to the border of the PSD ( = 0 and indicated by dashed line). Data in this figure is the same dataset as used in Fig. 3, as these figures show different aspects of the same experiment. Data are represented as means ± SEM. Source data are provided as a Source Data file.

periods of free and confined diffusion. Using a critical threshold of confinement $L_c$ we defined regions of confined diffusion, or confinement zones (Figs. 4c and S4f). mGluR5 mobility was strongly reduced inside these confinement zones (median $D_{eff}$ inside: 0.01 μm²/s and outside: 0.068 μm²/s; Fig. S4g, h). The average radius of confinement zones was 79.8 ± 0.97 nm, and receptors remained confined for 0.85 ± 0.03 s (Fig. 4d, e). Interestingly, we frequently observed that trajectories undergo confinement specifically in the perisynaptic zone

(Fig. 4f–h). Indeed, when we mapped the peaks of confinement zones and centers of the immobile trajectories, we detected clear hotspots of reduced mGluR5 mobility around synapses (Figs. 4i and S4i). To quantify this, we determined the distance of the confinement zones to the PSD border. Strikingly we found that the vast majority of confinement zones were located within the perisynaptic zone, <100 nm from the PSD (Fig. 4j). Similarly, immobile tracks were also enriched in the perisynaptic zone (Fig. 4k). Together, these experiment reveal that the

diffusion of mGluR5 around PSDs is highly heterogeneous and that mGluR5 is transiently confined primarily at perisynaptic zones, close to the border of the PSD.

## The C-terminal domain of mGluR5 mediates perisynaptic confinement

The particular heterogeneous organization of mGluR5 dynamics suggests that specific mechanisms retain the receptor in the perisynaptic zone. The large intracellular C-terminal domain (CTD) of mGluR5a contains many protein interaction motifs and phosphorylation sites involved in surface expression and trafficking[35]. However, whether the CTD of mGluR5 contributes to the spatial heterogeneity of surface mobility remains unknown. To test this, we generated a mutant lacking the last 314 C-terminal amino acids (SEP-mGluR5ΔC) (Fig. 5a). Truncation of the mGluR5 CTD did not impact surface expression (Fig. S5a), as has been previously shown[36]. We first used gSTED microscopy to assess the localization of surface-expressed mGluR5ΔC relative to the PSD. Compared to mGluR5 wild-type (mGluR5WT), mGluR5ΔC showed a similar exclusion from the PSD (Fig. 5b–d). However, we found a significantly increased fraction of spines with a homogeneous distribution of mGluR5ΔC (synaptic enrichment: WT: $11 \pm 2\%$ and ΔC: $7.1 \pm 2\%$, synaptic + perisynaptic enrichment: WT: $16 \pm 3\%$ and ΔC: $12 \pm 3\%$, perisynaptic enrichment: WT: $57 \pm 3\%$ and ΔC: $51 \pm 3\%$ and homogeneous distribution: WT: $16\% \pm 3$ and ΔC: $30\% \pm 5\%$; Fig. 5d). Furthermore, the loss of the CTD resulted in the loss of mGluR5 enrichment in spines (WT: $1.47 \pm 0.045$; ΔC: $1.08 \pm 0.032$; Figs. 5e and S5a).

To further investigate whether the CTD is involved in mediating mGluR5 confinement in the perisynaptic zone, we performed SMT. Significantly fewer mGluR5ΔC trajectories were found to be perisynaptic, and more tracks were only transiently associated with the perisynaptic zone (Fig. 5f). We also observed that mGluR5ΔC tracks were more homogeneously distributed (Fig. 5j). The diffusion coefficient was significantly increased for both perisynaptic and transient perisynaptic trajectories of mGluR5ΔC (median $D_{eff}$ perisynaptic: WT: $0.031 \, \mu m^2/s$ and ΔC: $0.056 \, \mu m^2/s$, median $D_{eff}$ transient perisynaptic: WT: $0.069 \, \mu m^2/s$ and ΔC: $0.096 \, \mu m^2/s$), but not of synaptic mGluR5ΔC trajectories (median $D_{eff}$ synaptic: WT: $0.020 \, \mu m^2/s$ and ΔC: $0.026 \, \mu m^2/s$), compared to mGluR5WT (Fig. 5g, j). Consistently, the fraction of immobile trajectories was significantly reduced for mGluR5ΔC (WT: $0.64 \pm 0.03$ and ΔC: $0.44 \pm 0.03$; Figs. 5k and S5b), as well as the fraction of mobile mGluR5ΔC trajectories with transient confinement zones (WT: $0.43 \pm 0.03$ and ΔC: $0.25 \pm 0.02$; Figs. 5l and S5c). Even when mGluR5ΔC was transiently confined, diffusion inside the confinement zones was significantly faster compared to mGluR5WT (median $D_{eff}$ WT: $0.0094 \, \mu m^2/s$ and ΔC: $0.015 \, \mu m^2/s$; Figs. 5h and S5d) and mGluR5ΔC confinement zones were on average larger (radius WT: $76.7 \pm 2.3 \, nm$ and ΔC: $99.4 \pm 2.3 \, nm$; Fig. S5e). We found that the mGluR5ΔC confinement zones were more homogeneously distributed and particularly showed less enrichment immediately adjacent to the PSD compared to mGluR5WT confinement zones (fraction confinement zones at 25 nm distance from PSD: WT: $0.20 \pm 0.03$ and ΔC: $0.14 \pm 0.03$; Fig. 5i). Consistently, the map of mGluR5ΔC confinement and immobility hotspots also revealed less pronounced areas of restricted mGluR5 diffusion in the perisynaptic zone (Fig. 5m). These results further indicate that the mGluR5 CTD contributes to the transient confinement of mGluR5 in perisynaptic nanodomains.

## The C-terminal domain of mGluR5 prevents synaptic entry

In stark contrast to AMPARs, mGluR5 seems to be transiently enriched in perisynaptic nanodomains, and almost completely excluded from the PSD. We, therefore, hypothesized that apart from mechanisms that confer perisynaptic retention of mGluR5, specific mechanisms prevent the synaptic entry of mGluR5. To begin to test this, we reasoned that we could target mGluR5 to the PSD by fusing mGluR5 to the CTD of

Stargazin (STGtail), the AMPAR auxiliary protein that associates with PSD-95 to concentrate AMPARs in the PSD (Fig. 6a)[37,38]. When we coupled the STGtail to a single transmembrane domain with an N-terminal SEP-tag (SEP-pDisp-STGtail), this construct was efficiently targeted to synapses marked by Homer1c-mCherry (Fig. S6a–c). Surprisingly, however, when mGluR5 was directly fused to the STGtail (mGluR5-STGtail), the receptor was still largely excluded from the PSD (Fig. 6b–d). For mGluR5-STGtail we observed a modest but significant increase in the number of spines with synaptic and perisynaptic enrichments (mGluR5WT: $16 \pm 3\%$ and mGluR5-STGtail: $29 \pm 3\%$; Fig. 6d), showing that the attempt to recruit mGluR5 to the PSD by the addition of the STGtail was only successful in a few spines. This also resulted in a reduction of spines with a homogeneous distribution (mGluR5WT: $16 \pm 3\%$ and mGluR5-STGtail: $5.3 \pm 2\%$; Fig. 6d). Overall, however, the distribution of mGluR5-STGtail was similar to mGluR5WT (Figs. 6b–d and 5b–d), corroborated by the unchanged enrichment in spines (mGluR5WT: $1.47 \pm 0.035$ and mGluR5-STGtail: $1.58 \pm 0.053$; Figs. 6e and S6d).

Considering that the mGluR5 CTD mediates perisynaptic confinement (Fig. 5), we predicted that it may also play a critical role in preventing the synaptic entry of mGluR5. To test this idea we made a chimera construct replacing the CTD of mGluR5 for the STGtail: SEP-mGluR5ΔC-STGtail (mGluR5ΔC-STGtail) (Fig. 6a). Interestingly, this mGluR5-Stargazin chimera was very efficiently recruited to the PSD, marked by Homer1c (Fig. 6b, c). We found a 5-fold increase in the percentage of spines with synaptic enrichment, compared to mGluR5WT, and a decrease in spines with a perisynaptic distribution (mGluR5ΔC-STGtail: synaptic enrichment: $64 \pm 5\%$, synaptic + perisynaptic enrichment: $15 \pm 2\%$, perisynaptic enrichment: $15 \pm 3\%$ and homogeneous distribution: $5.5 \pm 1\%$; Fig. 6d). Furthermore, we observed a significant increase in spine enrichment of mGluR5ΔC-STGtail (mGluR5ΔC-STGtail: $2.14 \pm 0.064$; Fig. 6e). Similarly, SMT of mGluR5-STGtail showed that the addition of the STGtail did not affect the distribution of receptor diffusion (Figs. 6f and 5j). However, in the few instances that mGluR5-STGtail entered the PSD, it was more immobile compared to mGluR5WT (median synaptic $D_{eff}$ mGluR5WT: $0.020 \, \mu m^2/s$, mGluR5-STGtail: $0.015 \, \mu m^2/s$ and mGluR5ΔC-STGtail: $0.014 \, \mu m^2/s$; Fig. S6e). In contrast, the confinement zones and immobile trajectories of mGluR5ΔC-STGtail were strongly enriched within the PSD (Fig. 6g–i). To ensure that the STGtail is properly exposed in the mGluR5-STGtail construct, we designed mGluR5-STGtail$_{split}$ where we positioned the mGluR5 CTD at amino acid position 302 in the STGtail, just upstream of the PDZ-binding motif (Fig. S6f). It has previously been validated that insertion of a fluorophore (GFP or mCherry), similar in size to the mGluR5 CTD, at position 302 in Stargazin results in a synaptically localized protein shown by co-localization with PSD-95[39]. In this construct, the putative membrane-bound RS domain (a stretch of seven arginines interleaved by nine serines) at the start of the STGtail is able to attach to the plasma membrane, without masking the PDZ-binding motif at the end. Interestingly, lengthening of the STGtail by an artificial linker after the RS domain (in addition to the fluorophore) has even been shown to potentiate binding to the lower PDZ domains of PSD-95 which is oriented perpendicularly to the plasma membrane[39] (Fig. S6f). mGluR5-STGtail$_{split}$ showed the mGluR5-typical perisynaptic localization in spines as shown with STED microscopy (Fig. S6g, h) and we observed no differences in spine enrichment compared to mGluR5-STGtail (1−mGluR5-STGtail$_{split}$: $1.77 \pm 0.11$, 2−mGluR5-STGtail: $1.72 \pm 0.10$ and 3−mGluR5ΔC-STGtail: $3.29 \pm 0.23$; Fig. S6l). To assess whether the mGluR5 CTD is sufficient for the synaptic exclusion of the STGtail, we coupled the mGluR5 CTD and STGtail$_{split}$ to a single transmembrane domain (pDisp-CTD-STGtail$_{split}$; Fig. S6i). Indeed, we observed that the spine enrichment was significantly reduced in pDisp-CTD-STGtail$_{split}$ compared to pDisp-STGtail (pDisp-CTD-STGtail$_{split}$: $1.87 \pm 0.25$ and pDisp-STGtail: $7.6 \pm 0.67$; Fig. S6i) with preferential

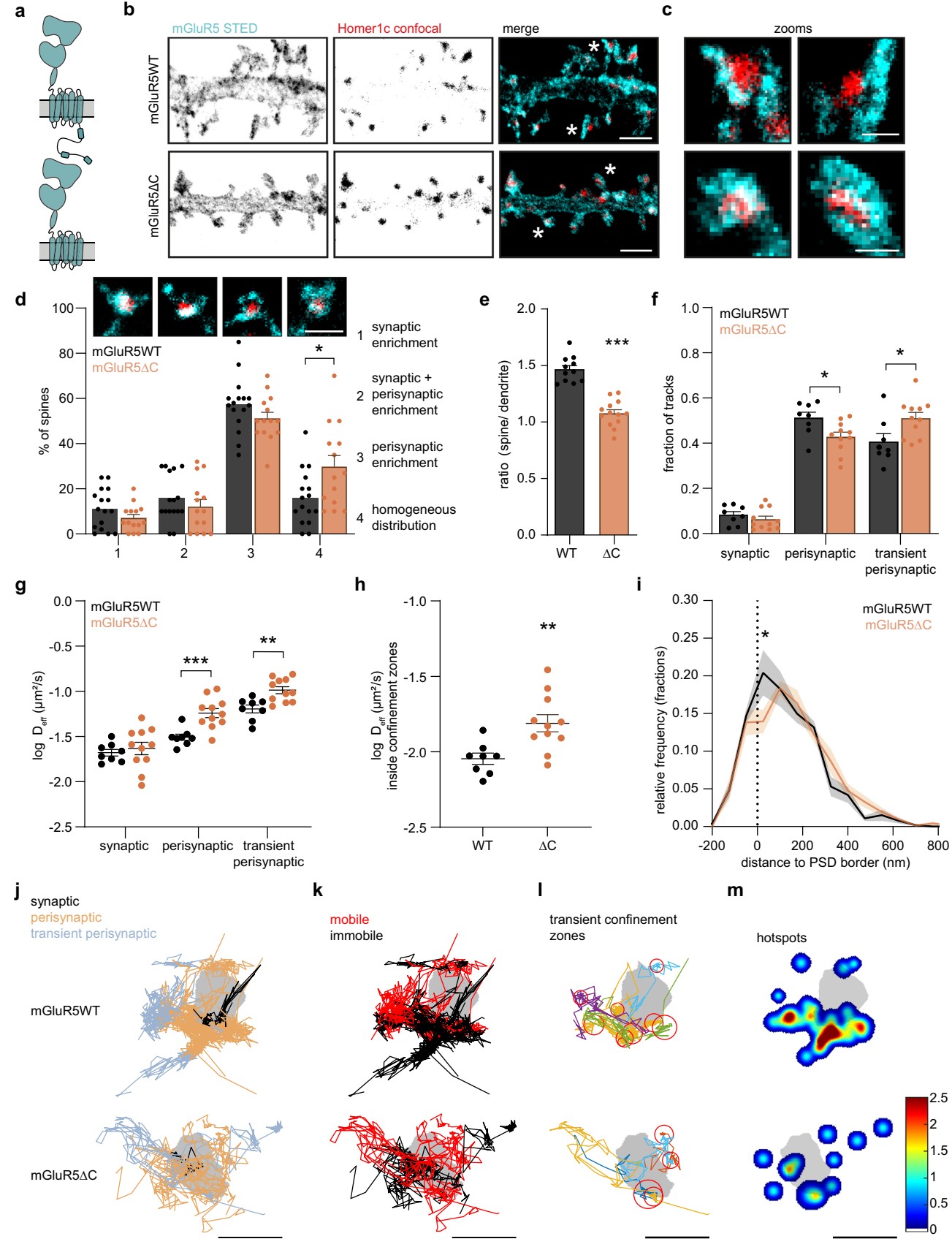

**Fig. 5 | The C-terminal domain of mGluR5 mediates perisynaptic confinement.**
**a** Schematic of monomeric full-length mGluR5WT (top) and mGluR5ΔC (bottom) lacking its C-terminal tail. **b** Representative gSTED images of dendrite expressing SEP-mGluR5WT and SEP-mGluR5ΔC, additionally labeled with an anti-GFP nanobody Atto647N (cyan), and Homer1c-mCherry (red; confocal). Scale bar, 2 μm.
**c** Zooms of spines indicated in **b** with asterisks. Scale bar, 500 nm. **d** Quantification of mGluR5WT (black; $n = 16$) and mGluR5ΔC (orange; $n = 14$) localization in spines: (1) synaptic enrichment ($p = 0.1565$), (2) synaptic + perisynaptic enrichment ($p = 0.1488$), (3) perisynaptic enrichment (0.3696) and (4) homogeneous distribution ($p = 0.0204$; two-sided unpaired $t$ test for each category). On top are representative images of the different categories of mGluR5 localization (cyan), relative to Homer1c (red), at spines. Scale bar, 1 μm. **e** Quantification of the ratio of spine over dendrite intensity of mGluR5WT ($n = 11$) and mGluR5ΔC ($n = 13$, $p < 0.0001$; unpaired $t$ test). **f** Fraction of synaptic ($p = 0.3529$), perisynaptic ($p = 0.0218$) and transient perisynaptic trajectories ($p = 0.0254$) of mGluR5WT ($n = 8$) and mGluR5ΔC ($n = 11$; two-sided unpaired $t$ test for each category). **g** Mean log $D_{eff}$ per

neuron of synaptic ($p = 0.5923$), perisynaptic ($p = 0.0008$), and transient perisynaptic ($p = 0.0025$) trajectories of mGluR5WT ($n = 8$) and mGluR5ΔC ($n = 11$; two-sided unpaired $t$ test for each category). **h** Mean log $D_{eff}$ per neuron of trajectories inside confinement zones of mGluR5WT ($n = 8$) and mGluR5ΔC ($n = 11$, $p = 0.0053$; two-sided unpaired $t$ test). **i** Relative frequency distribution of the distance of confinement zones of mGluR5WT ($n = 8$) and mGluR5ΔC ($n = 11$) to the border of the PSD (=0 and indicated by dashed line) (two-way repeated measures ANOVA with Bonferroni's multiple comparisons test, at 25 nm distance from PSD border: $p = 0.0003$). **j** Example synapses of mGluR5WT and mGluR5ΔC with trajectories color-coded for being synaptic (black), perisynaptic (orange), and transient perisynaptic (blue), **k** for being mobile (red) and immobile (black), **l** for being transiently confined trajectories (random colors) with corresponding confinement zones (red circles), and **m** hotspots of immobile tracks (shown in **k**) and confinement zones (shown in **l**), color-coded for their frequency. Scale bars, 500 nm. Data are represented as means ± SEM. *$p < 0.05$, **$p < 0.01$ and ***$p < 0.001$. Source data are provided as a Source Data file.

---

perisynaptic localization in spines (Fig. S6j, k). However, even though pDisp-CTD-STGtail$_{split}$ was significantly less enriched in spines compared to mGluR5ΔC-STGtail, some neurons displayed a similar spine enrichment. These results show that removing the mGluR5 CTD and increasing the affinity of mGluR5 for the PSD allow the entry and retention of mGluR5 in synapses, indicating that the mGluR5 CTD regulates both the retention and synaptic exclusion of mGluR5.

### Inducible heterodimerization system allows robust and rapid recruitment of mGluR5 to the synapse

We hypothesized that the distinct segregation of ionotropic and metabotropic glutamate receptor types in different subsynaptic domains optimizes synaptic signaling. To better understand the functional relevance of mGluR5 nanodomains in the perisynaptic zone, we set out to develop a system to acutely control mGluR5 distribution to study the effect of mGluR5 positioning on synaptic signaling. To do so, we used the inducible FKBP-rapalog-FRB heterodimerization system, a reliable and robust tool to induce interactions between two proteins by the addition of rapalog[40]. To allow controlled recruitment of mGluR5 to the synaptic scaffold Homer1c we developed FRB-tagged mGluR5 and FKBP-tagged Homer1c constructs (Fig. 7a). Indeed, the enrichment of mGluR5 in spines significantly increased upon the addition of rapalog (before: $1.5 \pm 0.08$ and 50 min after: $2.7 \pm 0.2$; Fig. S7a–c). Importantly, Homer1c spine enrichment was not different between neurons incubated with rapalog and control neurons where a vehicle was added, indicating unidirectional recruitment of mGluR5 towards Homer1c which is stably retained in the PSD. Live-cell imaging further demonstrated that mGluR5 accumulated within synapses over the time course of 40 min and we observed a clear re-distribution of mGluR5 into the PSD (Fig. 7b–f). Together, these data show that this rapalog-inducible system can be employed to acutely and robustly relocate mGluR5 to postsynaptic sites.

### Synaptic recruitment of mGluR5 alters synaptic signaling

Activation of postsynaptic mGluRs modulates spine $Ca^{2+}$ levels via several routes: via IP3-sensitive intracellular stores[14], modulation of voltage-gated calcium channels (VGCCs), NMDA receptors, or $Ca^{2+}$-induced $Ca^{2+}$ release (CICR)[41–44]. However, direct measurements of the contribution of perisynaptic mGluR5 to synaptic calcium signaling at the level of a single synapse is difficult, and conclusions thus far rely on biophysical models[17]. Nevertheless, it has been generally assumed that spontaneous glutamate release events would not activate mGluR5. If the contribution of mGluR5 to synaptic calcium levels are minimal during spontaneous release events due to its perisynaptic localization, the calcium events should drastically change upon recruiting mGluR5 to the center of synapses, also experimentally revealing the significance of its perisynaptic localization during spontaneous release. To study the dynamic changes in spine calcium concentrations we

used the optical $Ca^{2+}$ sensor GCaMP6f[45]. We expressed GCaMP6f and imaged neurons at DIV21-23 in extracellular buffer containing 3 μM TTX and 0 mM $Mg^{2+}$ to block action potentials and relieve the NMDA receptor pore block. GCaMP6f robustly reported miniature spontaneous $Ca^{2+}$ transients (mSCTs) that were detected in individual dendritic spines without detected $Ca^{2+}$ increases in the dendritic shaft or neighboring spines (Fig. S8a–c), consistent with previous studies[46,47]. We found a broad range of event frequencies per neuron, ranging from 0 to 25 events/50 seconds, with 90.6% of neurons exhibiting at least one event per 50 seconds. Moreover, the mSCT frequency was significantly increased by a 5-min application of the group I mGluR specific agonist DHPG (median mSCT frequency: basal: 0.30 Hz, 95% CI [0.14 0.68], DHPG: 0.78 Hz, 95% CI [0.40 1.86] and AP5: 0.06 Hz, 95% CI [0.00 0.10]; Fig. S8d–f), confirming that endogenous mGluR5 contributes to synaptic calcium signaling. Treatment with the NMDAR antagonist AP5 eliminated most events, indicating that activation of mGluR5, at least in part, induces spine mSCTs by potentiating NMDARs (Fig. S8d–f).

Next, to investigate the spatiotemporal effects of mGluR distribution on synaptic function at individual synapses, we combined the inducible FKBP-rapalog-FRB heterodimerization system and GCaMP6f and determined the effect of mGluR5 recruitment to the synapse on synaptic calcium signaling. We co-expressed GCaMP6f with SNAP-mGluR5-FRB and FBKP-Homer1c-mCherry and imaged GCAMP6f before and 30 min after the application of rapalog. In the maximum intensity projections of the obtained GCaMP6f streams (50 ms) we observed a clear increase in peak intensities at individual spines after a 30-min rapalog incubation (Fig. 8a–c). Indeed, quantification consistently showed that rapalog application caused a dramatic threefold increase in mSCT frequency (median mSCT frequency: before: 0.10 Hz, 95% CI [0.06 0.18] and after: 0.32 Hz, 95% CI [0.20 0.48]; Fig. 8d and S8h), also when corrected for spine density (Fig. S8g). The mSCT amplitude was not changed after the addition of rapalog (median $\Delta F/F_0$: before: 0.057, 95% CI [0.052 0.066] and after: 0.059, 95% CI [0.055 0.064]; Fig. S8i), but we did find significantly larger decay tau times (before: 0.13 s, 95% CI [0.12 0.15] and after 0.17 s, 95% CI [0.14 0.23]; Fig. 8f, g). To control for possible undesired side effects of rapalog on mSCT frequency and amplitude we performed the same experiment but with mGluR5 lacking the FRB domain. In this experiment, we observed no differences in mSCT frequency (median mSCT frequency: before: 0.06 Hz, 95% CI [0.02 0.12] and after: 0.04 Hz, 95% CI [0.02 0.14]; Figs. 8e and S8h) and amplitude (median $\Delta F/F_0$: before: 0.054, 95% CI [0.045 0.064] and after: 0.054, 95% CI [0.048 0.062]; Fig. S8j) before and after rapalog application. Altogether, these data support the model that positioning mGluR5 at perisynaptic sites is critical to restrict mGluR5

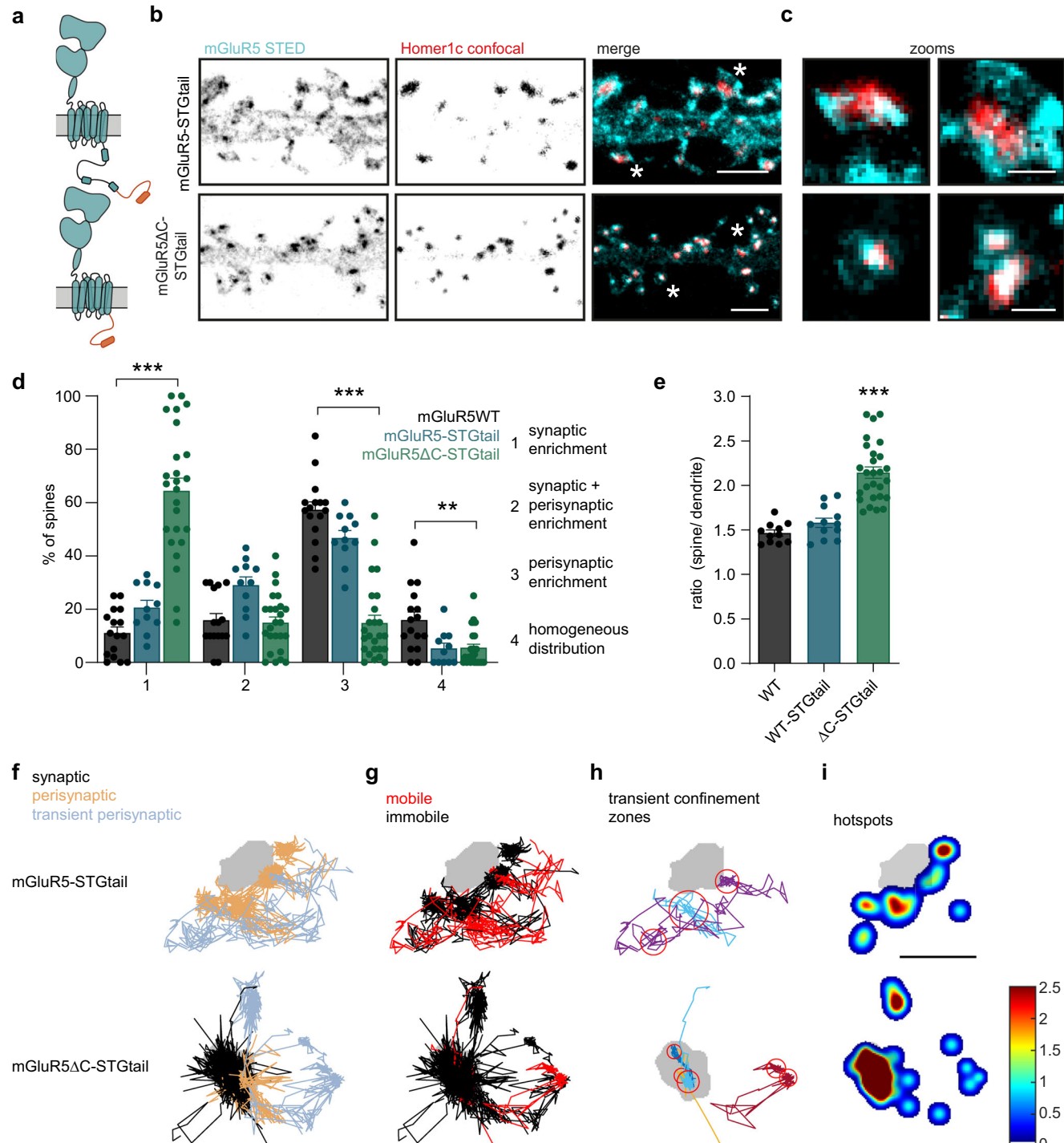

**Fig. 6 | The C-terminal domain of mGluR5 prevents synaptic entry. a** Schematic of mGluR5-STGtail (top) and mGluR5ΔC-STGtail (bottom). **b** Representative gSTED images of dendrite expressing SEP-mGluR5-STGtail and SEP-mGluR5ΔC-STGtail, additionally labeled with an anti-GFP nanobody Atto647N (cyan), and Homer1c-mCherry (red; confocal). Scale bar, 2 μm. **c** Zooms of spines indicated in **b** with asterisks. Scale bar, 500 nm. **d** Quantification of mGluR5WT (black; *n* = 16), mGluR5-STGtail (blue; *n* = 11), and mGluR5ΔC-STGtail (green; *n* = 25) localization in spines: (1) synaptic enrichment, (2) synaptic + perisynaptic enrichment, (3) perisynaptic enrichment and (4) homogeneous distribution (*p* < 0.0001, Kruskal–Wallis test for each category with Dunn's multiple comparisons test: *p* = 0.5398 for mGluR5WT vs. mGluR5-STGtail, *p* < 0.0001 for mGluR5WT vs. mGluR5ΔC-STGtail and *p* = 0.0011 for mGluR5-STGtail vs. mGluR5ΔC-STGtail). **e** Quantification of the ratio of spine over dendrite intensity of mGluR5WT (*n* = 11), mGluR5-STGtail (*n* = 12) and

mGluR5ΔC-STGtail (*n* = 27; *p* < 0.0001, one-way ANOVA with Dunnet's multiple comparisons test: compared to mGluR5WT *p* = 0.4700 for mGluR5-STGtail and *p* < 0.0001 for mGluR5ΔC-STGtail). **f** Example synapses of mGluR5-STGtail and mGluR5ΔC-STGtail with trajectories color-coded for being synaptic (black), perisynaptic (orange), and transient perisynaptic (blue), **g** for being mobile (red) and immobile (black), **h** for being transiently confined trajectories (random colors) with corresponding confinement zones (red circles), and **i** hotspots of immobile tracks (shown in **g**) and confinement zones (shown in **h**), color-coded for their frequency. Scale bar, 500 nm. The mGluR5WT dataset shown in **d** and **e** is also shown in Fig. 5d, e, as these figures show different aspects of the same experiment. Data are represented as means ± SEM. *\*p* < 0.05, \*\**p* < 0.01, and \*\*\**p* < 0.001. Source data are provided as a Source Data file.

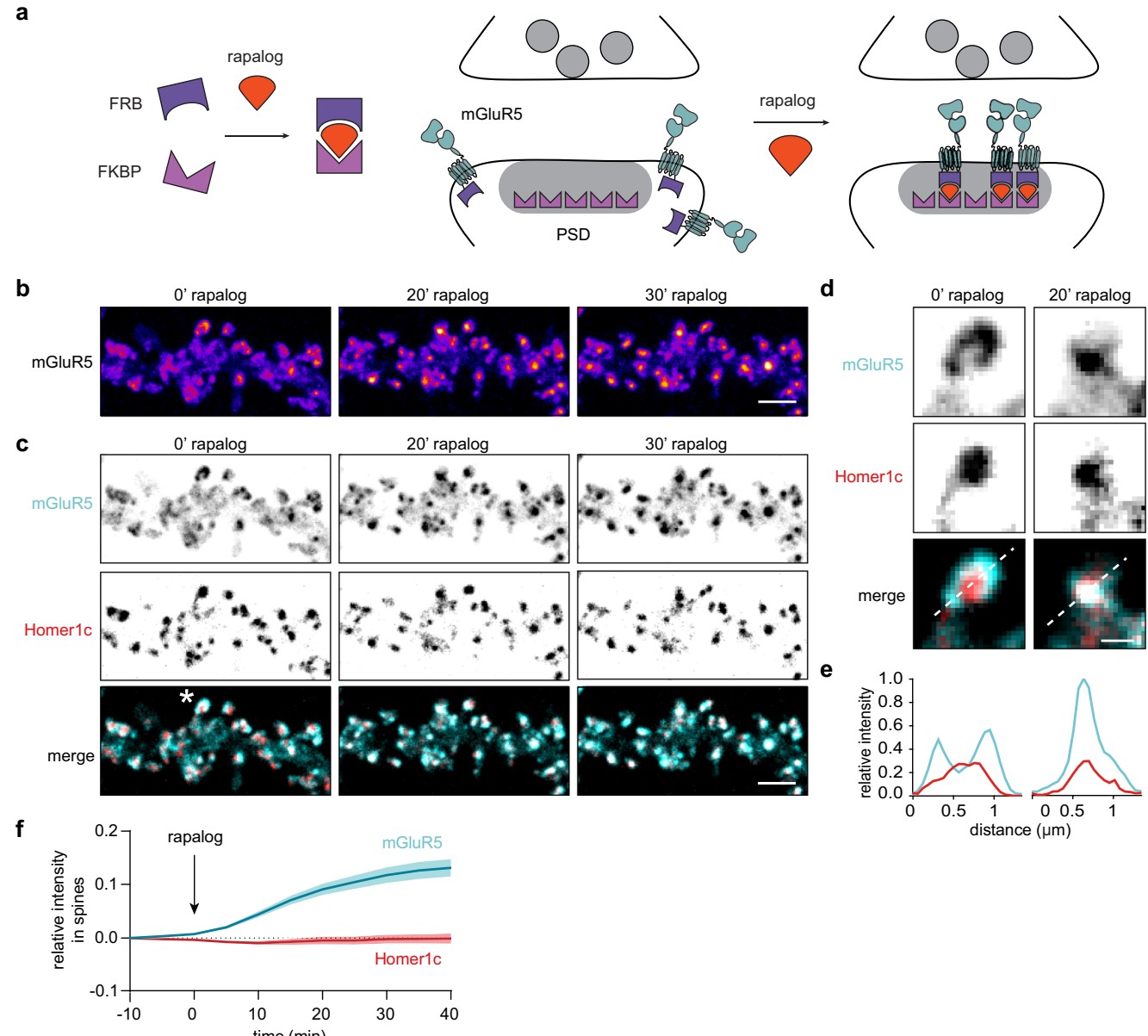

**Fig. 7 | mGluR5 is efficiently recruited to the synapse using the inducible FKBP-rapalog-FRB heterodimerization system. a** Schematic of the FKBP-rapalog-FRB heterodimerization system (left) and how this system is used to recruit mGluR5 to the PSD (right). **b** Live-cell time-lapse images of SEP-mGluR5-FRB before (0') and 20 and 30 min after rapalog application. The dendrites are color-coded for the fluorescence intensity of SEP-mGluR5-FRB. Scale bar, 2 µm. **c** Live-cell time-lapse images of the same dendrite as shown in **b** showing the relative localization SEP-mGluR5-FRB (cyan) and 2xFKBP-Homer1c-mCherry (red) before (0') and 20 and 30 min after rapalog application. Scale bar, 2 µm. **d** Zoom of spine indicated in **c** with asterisk before (0') and 20 min after rapalog application. Scale bar, 500 nm. **e** Line profile of the spine in **d**, indicated with dotted line, showing the localization of mGluR5 (cyan) relative to Homer1c (red) before (0') and 20 min after rapalog application. **f** Quantification of SEP-mGluR5-FRB (cyan) and 2xFKBP-Homer1c-mCherry (red) intensity in spines over time upon rapalog application at $t = 0$ ($n = 17$). Data are represented as means ± SEM. Source data are provided as a Source Data file.

overactivation during spontaneous release events as the acute recruitment of mGluR5 to the synapse results in aberrant synaptic calcium signaling (Fig. 8h).

## Discussion

The subsynaptic organization of group I mGluRs modulates their activation and subsequent downstream signaling, essential for proper synaptic transmission and plasticity. However, fundamental aspects of mGluR distribution and dynamics at excitatory synapses are still poorly understood. Here, we present a mechanistic understanding of how the CTD of mGluR5 controls its dynamic organization in perisynaptic nanodomains, as well as prevents mGluR5 from

entering the synapse, allowing mGluR5 to finely tune synaptic calcium signaling.

Our localization and SMT data show that mGluR5 is enriched in the perisynaptic zone and largely absent from the PSD, consistent with early EM studies[11–13] and recent super-resolution microscopy studies[48]. Importantly, we observed that the organization of mGluR5 is much more heterogeneous than suggested before. We found that mGluR5 assembles in distinct perisynaptic nanodomains, suggesting that specific mechanisms hinder mGluR5 diffusion at the perisynaptic zone. Consistently, our SMT data revealed that mGluR5 trajectories were enriched in the perisynaptic zone and were confined to domains with radii ranging from 40 up to 200 nm. Interestingly, however, mGluR5

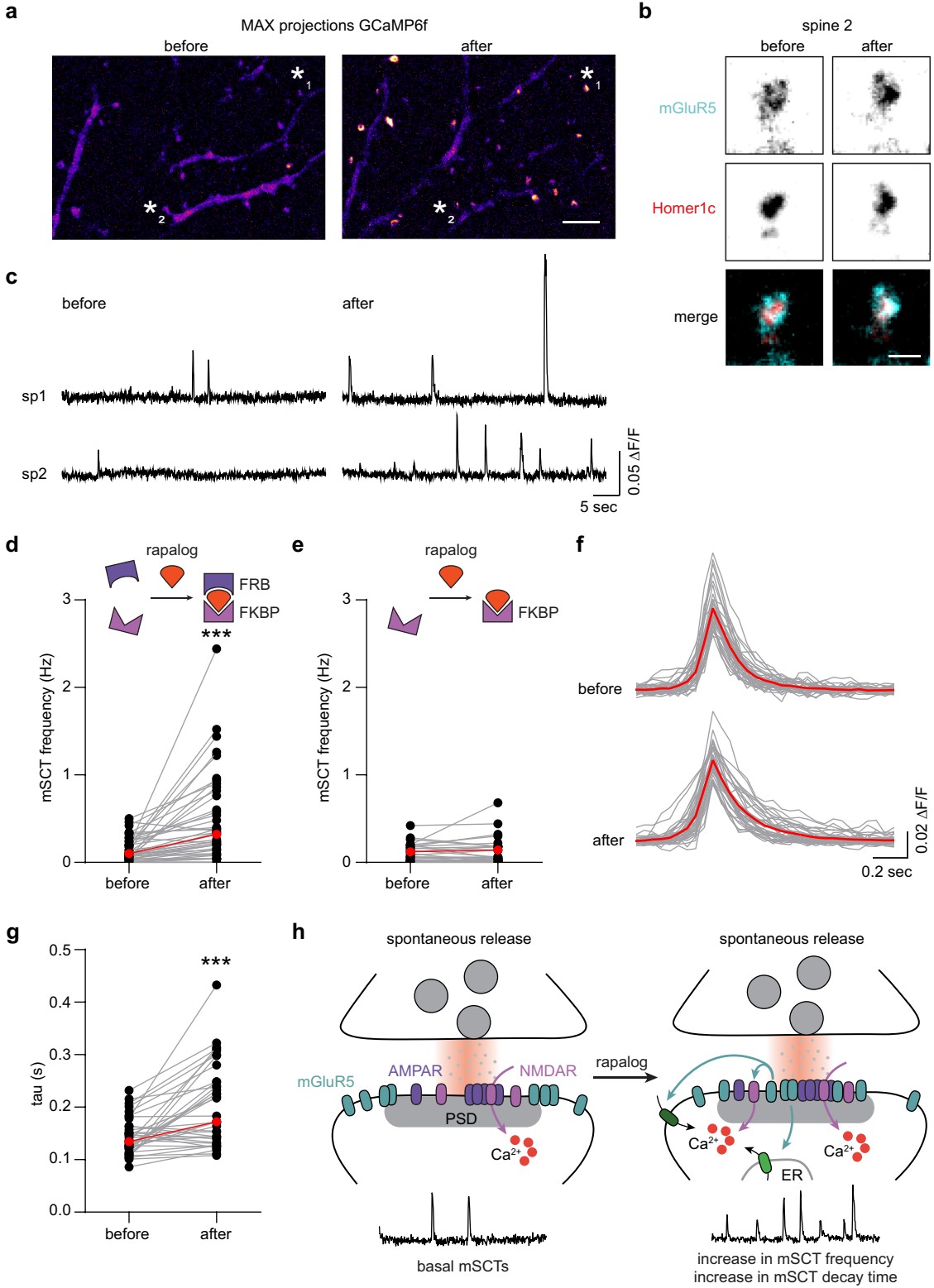

was only transiently trapped in these perisynaptic nanodomains, and rapidly exchanged between diffusive and confined states. Only a small fraction of mGluR5 seemed to be retained in the PSD, possibly by indirect steric hindrance, or molecular crowding mechanisms[49,50]. Thus, the enrichment of mGluR5 in perisynaptic nanodomains is the result of a highly dynamic equilibrium of diffusion states.

The transient confinement of mGluR5 may represent either binding and unbinding to an interaction partner or hindrance of movement of mGluRs due to other mechanisms. We found that the perisynaptic retention of mGluR5, but also the exclusion from the synapse is largely controlled by its CTD. Most importantly, the removal of the mGluR5 CTD resulted in a higher mobility in the perisynaptic zone and less transient confinement in perisynaptic nanodomains. Based on these data, we propose that the mGluR5 CTD is critical for the transient confinement of mGluR5 in perisynaptic nanodomains, possibly through stabilizing interactions at the perisynaptic zone. The CTD

**Fig. 8 | Synaptic recruitment of mGluR5 increases the frequency of spontaneous synaptic Ca²⁺ transients. a** Maximum projections of the GCaMP6f stream (50 s) in a representative dendrite before (baseline) and after 30 min rapalog application. Scale bar, 5 μm. **b** Zoom of spine 2 indicated in *a* with asterisk, expressing SNAP-mGluR5-FRB labeled with the cell-impermeable SNAPdye JF646 (cyan) and 2xFKBP-Homer1c-mCherry (red) before and 30 min after rapalog-induced recruitment. Scale bar, 1 μm. **c** ΔF/F₀ traces of GCaMP6f signal from two spines indicated in **a** with asterisks before and after 30 min of rapalog-induced recruitment of mGluR5 to Homer1c. **d** Quantification of mSCT frequencies upon application of rapalog in neurons expressing SNAP-mGluR5-FRB and 2xFKBP-Homer1c-mCherry ($n = 43$ neurons, $p < 0.0001$, two-sided Wilcoxon matched-pairs signed rank test) and **e** in neurons expressing SNAP-mGluR5 and 2xFKBP-Homer1c-mCherry (control; $n = 37$ neurons, $p = 0.5819$, two-sided Wilcoxon matched-pairs signed rank test). **f** Average traces of all mSCTs per neuron (gray) and average mSCT trace of all neurons (red) before and after 30 min of rapalog. **g** Quantification of mSCT decay tau times (s) upon application of rapalog ($n = 37$ neurons, $p < 0.0001$, two-sided Wilcoxon matched-pairs signed rank test). **h** Model of deregulated calcium signaling upon mGluR5 recruitment to the synapse during spontaneous synaptic activity. Medians are indicated by the red lines. Source data are provided as a Source Data file.

---

of mGluRs can interact with a variety of intracellular proteins, including the scaffolding protein Homer1b/c that links mGluRs to a larger synaptic complex[35,51–54]. Numerous studies have proposed Homer1b/c as the protein regulating the subsynaptic positioning of mGluRs[55–58]. We found however that Homer1c overexpression did not affect mGluR5 enrichment in spines nor did it affect mGluR5 diffusion. In fact, we found that mGluR5 localizes away from Homer1c, being present in the core of the PSD[59], and that forced recruitment of mGluR5 to Homer1c using the FKBP-heterodimerization-FRB system is required to recruit mGluR5 to the PSD. This might suggest that mGluR5 only interacts with Homer1c molecules present at the periphery of the PSD. The mGluR5 CTD also contains many other binding motifs and phosphorylation sites that might underlie the dynamic positioning of mGluRs[3,35]. Also, other mechanisms such as phase separation, molecular crowding, lipid organization, or cytoskeletal hindrance might mediate the organizational properties of the mGluR5 CTD[2,60]. Furthermore, we cannot exclude the possibility that other mGluR5 domains, including the extracellular N-terminal domain, are involved in receptor positioning[19,61]. For example, the enrichment of AMPARs at synaptic sites has been ascribed both to the CTD[62–64], as well as the NTD[65–67]. Based on these findings we propose that at excitatory synapses functionally distinct glutamate receptor types are spatially segregated in subsynaptic domains, in part via intracellular interactions that can either promote or hinder the entry of receptors into the PSD. Our findings reveal that postsynaptic mGluR positioning is regulated by conceptually novel mechanisms that effectively retain mGluR5 close to the synapse, but segregated away from the core synaptic membrane, to efficiently modulate synaptic function.

The preferential perisynaptic organization we observed for mGluR5 is likely to affect receptor activation and function. The perisynaptic mGluRs are perfectly situated to detect glutamate spillover from the synaptic cleft during sustained or high-frequency stimulation and initiate downstream signaling. Furthermore, the perisynaptic nanodomains may function to concentrate signaling machineries optimizing the ability of mGluRs to connect to downstream signaling effectors. Such local accumulations of receptors and their effectors, in so-called signalosomes, have been shown to contribute to the efficiency and fidelity of signal transmission[68,69]. In support of this concept of perisynaptic signalosomes, the mGluR5 downstream signaling partners Gαq/Gα11, PLCβ, DGL-α, and Norbin were found to closely parallel the organization of mGluR5 as they were either enriched in the perisynaptic zone or found to colocalize with mGluR5[70–74]. In general, we observed that mGluR5 was not limited to one perisynaptic nanodomain, but formed multiple distinct domains in the perisynaptic zone. This opens the intriguing possibility that mGluRs assemble into distinct signalosomes that each consist of a specific subset of signaling molecules. The compartmentalization of downstream effectors of mGluRs might be of critical importance to regulate the initiation of downstream signaling and warrant the functional selectivity of mGluR1 and mGluR5. The perisynaptic zone also contains a stable endocytic zone (EZ) that functions to locally internalize and recycle synaptic receptors[28,75–77]. In particular, the tight coupling of the PSD to the EZ has been shown to govern the efficient trafficking of mGluR5, regulating mGluR5 surface expression and signaling[78]. mGluR5 organized

in perisynaptic nanodomains that localize in close vicinity of the EZ might be particularly well-suited for fast desensitization and local endocytosis and recycling after activation to rapidly respond to sustained synaptic activity. Interestingly, mGluR5 is not exclusively present at perisynaptic sites. A small fraction of mGluR5 does localize to the PSD, which has been shown to be an activity-driven process to remodel mGluR5-scaffold interactions and modulate downstream signaling[79,80]. Also, mGluR5 broadly localizes throughout the dendritic shaft and, in contrast to excitatory synapses, localizes inside inhibitory synapses[81,82]. This heterogeneous localization suggests that mGluR5 dynamics are regulated by different processes in space and time to specify and support different mGluR5 functions. We found that recruiting mGluR5 from the perisynaptic zone to the core of the PSD strikingly increased calcium events at synapses. These results indicate that increasing the availability of mGluR5 in the PSD for activation during spontaneous synaptic activity strongly deregulates synaptic calcium signaling. The increased mSCT frequency upon recruitment of mGluR5 to the synapse argues for a direct (Ca²⁺ influx through NMDARs) or indirect (downstream activation of Ca²⁺ release) contribution of mGluR5 to mSCTs. Even though increased mSCT frequency is expected to increase NMDA current magnitudes, our results revealed no significant difference in mSCT amplitude indicating that mSCTs measured by GCaMP6f were not solely dependent on NMDA receptor activity. The increased mSCT frequency and decay times might reflect increased Ca²⁺ release from intracellular stores further shaping the mSCTs. Also, the number of mSCTs that were detectable and met by our detection criteria might be increased by Ca²⁺ release from internal stores (see M&M for details). Moreover, even though we performed the experiments in the absence of Mg²⁺, the Ca²⁺ influx through NMDARs might not always be sufficient to generate mSCTs and further relies on activation of Ca²⁺ release from internal stores to amplify NMDA-mediated Ca²⁺ transients[46]. We can also not exclude the possibility that other sources of Ca²⁺ entry, such as via VGCCs, also play a role here. It is nevertheless tempting to speculate that the induced synaptic enrichment of mGluR5 might to some extent induce the reported physical association between NMDA and mGluR5 initiated upon the activity-induced increase in Homer1a expression[24,80,83].

Altogether, our data provide an unforeseen level of mechanistic understanding of how postsynaptic mGluRs are transiently retained in distinct perisynaptic nanodomains to control synaptic signaling. Further delineation of these mechanisms will shed new light on how glutamatergic signaling is regulated by the cooperative actions of different glutamate receptor subtypes. The functional implications of erroneous mGluR positioning further underlines the relevance of understanding the relation between mGluR trafficking and signaling in the context of cognitive functioning.

## Methods

### Animals

All animal (male and female) experiments were performed in compliance with the guidelines for the welfare of experimental animals issued by the Government of the Netherlands (Wet op de Dierproeven, 1996) and European regulations (Guideline 86/609/EEC). All animal experiments were approved by the Dutch Animal Experiments Review

Committee (Dier Experimenten Commissie; DEC), and performed in line with the institutional guidelines of Utrecht University.

## Primary neuronal cultures and transfections

Hippocampal cultures were prepared from embryonic day 18 (E18) Janvier Wistar rat brains (both genders)[84]. Dissociated neurons were plated on coverslips coated with poly-L-lysine (37.5 μg/ml, Sigma-Aldrich) and laminin (1.25 μg/ml, Roche Diagnostics) at a density of 100,000 neurons per well in a 12-well plate. Neurons were grown in Neurobasal medium (NB) supplemented with 2% B27 (GIBCO), 0.5 mM glutamine (GIBCO), 15.6 μM glutamate (Sigma-Aldrich), and 1% penicillin/streptomycin (GIBCO) at 37 °C in 5% $CO_2$. Once per week, starting at 1 day in vitro (DIV1), half of the medium was refreshed with Brain-Phys Neuronal Medium (BP, STEMCELL Technologies), supplemented with 2% NeuroCult SM1 (STEMCELL Technologies) and 1% penicillin/streptomycin (GIBCO). At DIV3 (knock-in construct) or DIV11-16 neurons were transfected with indicated constructs using Lipofectamine 2000 (Invitrogen). Before transfection 300 μl conditioned medium was transferred to a new culture plate. For each well, 1.8 μg DNA was mixed with 3.3 μl Lipofectamine 2000 in 200 μl BP, incubated for 30 min at room temperature (RT), and added to the neurons. After 1–2 h, neurons were briefly washed with BP and transferred to the new culture plate with conditioned medium supplemented with an additional 400 μl BP with SM1 and penicillin/streptomycin and kept at 37 °C in 5% $CO_2$. All experiments were performed using neurons at DIV18–22. If neurons were kept longer than 6 days, medium was refreshed as described above.

## DNA constructs

The pRK5-SEP-mGluR5a and pRK5-myc-mGluR5a are previously described[78] and used as a template to make pRK5-SNAP-mGluR5a. To make pRK5-SEP-mGluR5aΔC, primers were designed using Gibson Assembly (NEBuilder HiFi DNA assembly cloning kit) to remove the last 314 C-terminal amino acids of mGluR5a. The pDisp-SEP-TM-STGtail construct was a gift from Dr. Thomas A. Blanpied[50] and the STGtail sequence was used to add to the C-terminal part of mGluR5a and mGluR5aΔC (before the stop codon) to make pRK5-SEP-mGluR5a-STGtail and pRK5-SEP-mGluR5aΔC -STGtail. Primers were designed to split the STGtail into two parts, STGtail203-302 and STGtail303-323, to make the pRK5-SEP-mGluR5-STGtail$_{split}$ and pRK5-SEP-pDisp-CTD-STGtail$_{split}$ constructs (as shown in Fig. S6f and i). The FKBP and FRB containing expression plasmids were a gift from Lukas C. Kapitein[85] and used to make pRK5-SNAP-mGluR5a-FRB, pRK5-SEP-mGluR5a-FRB, and 2xFKBP-Homer1c-mCherry using Gibson Assembly. Homer1c-mCherry and pSM155-mCherry have been described before[78]. The pRK5-SEP-mGluR5a was used as a template to replace mGluR5a with GluA2 (flip, Q/R edited), previously described in ref. [5]. GCaMP6f was a gift from Adam Cohen (Addgene plasmid # 58514) and PSD$_{FingR}$-mEos3.2 was a gift from Matthew J. Kennedy[86] and is based on (Dr. Don Arnold, Addgene plasmid # 46295)[26]. The pAAV-GFP-mGluR5 CRISPR/Cas9 ORANGE knock-in construct was designed as described in ref. [9]. The GFP-tag was inserted into the *Grm5* gene using the following target sequence: 5′–GTGCACAGTCCAGTGAGAGG–3′, resulting in the N-terminal tagging of mGluR5.

## Antibody and nanobody labeling

Neurons were fixed between DIV18-21 with 4% paraformaldehyde (PFA) and 4% sucrose in PBS for 10 min at RT, washed three times with PBS supplemented with 100 mM glycine (PBS/Gly), and blocked in 1–2% BSA in PBS/Gly for 30 min at RT. To label the surface-expressed pool of receptors, neurons were labeled with the GFP-booster Atto647N (Chromotek) diluted 1:200 in 1% BSA in PBS/Gly for 2 h at RT or Fluotag-X4 anti-GFP Atto647N (Nanotag) diluted 1:250 in 1% BSA in PBS/Gly for 1 h at RT. Neurons were then washed three times with PBS/

Gly, mounted in Mowiol mounting medium (Sigma), and imaged on the Leica SP8 microscope as described below.

For the GFP-mGluR5 knock-in experiments (Fig. 1i–m) neurons were fixed as described above and blocked with 10% NGS in PBS/Gly for 30 min at RT. To label, the surface-expressed receptors neurons were incubated with rabbit anti-GFP (MBL) diluted 1:2000 in 5% NGS in PBS/Gly for 2 h at RT and washed three times with PBS/Gly. Then, to label intracellular PSD-95, neurons were permeabilized in 0.25% Triton X-100 and 5% NGS in PBS/Gly for 10 min at RT, and incubated with mouse anti-PSD-95 (Neuromab) diluted 1:300 in 0.1% Triton X-100 and 5% NGS in PBS/Gly for 2 h at RT or overnight (O/N) at 4 °C. Neurons were washed three times and incubated with anti-rabbit Alexa488 and anti-mouse Alexa594 (Thermo Fisher Scientific) diluted 1:250 in 0.1% Triton X-100 and 5% NGS in PBS/Gly for 1 h at RT. Neurons were washed three times with PBS/Gly, mounted in Mowiol mounting medium (Sigma), and imaged on the Leica SP8 microscope as described below.

For two-color gSTED of SEP-mGluR5 and endogenous PSD-95 (Fig. S1a–d), neurons were fixed and labeled as described above, except for the secondary antibodies used. Rabbit anti-GFP and mouse anti-PSD-95 were visualized with anti-rabbit Atto647N (Sigma-Aldrich) and anti-mouse (Thermo Fisher Scientific) diluted 1:250.

For confocal imaging of mGluR5 expression levels in SEP-mGluR5 transfected and untransfected neurons (Fig. S1e–g), neurons were fixed as described above. Then neurons were blocked in 10% NGS and 0.1% Triton X-100 in PBS/Gly for 30 min at RT, incubated with rabbit anti-mGluR5 (Millipore) diluted 1:500 in 0.1% Triton X-100 and 5% NGS in PBS/Gly O/N at 4 °C. Neurons were washed three times and incubated with anti-rabbit Alexa594 (Thermo Fisher Scientific) diluted 1:250 in 0.1% Triton X-100 and 5% NGS in PBS/Gly for 1 h at RT.

For two-color gSTED of endogenous mGluR5 and PSD-95 (Fig. S1h–k) neurons were fixed and blocked as described above. Then neurons were, incubated with rabbit anti-mGluR5 (Millipore) diluted 1:500 and mouse anti-PSD-95 (Neuromab) diluted 1:300 in 0.1% Triton X-100 and 5% NGS in PBS/Gly O/N at 4 °C. Neurons were washed three times and incubated with anti-rabbit Atto647N (Sigma-Aldrich) and anti-mouse Alexa594 (Thermo Fisher Scientific) diluted 1:250 in 0.1% Triton X-100 and 5% NGS in PBS/Gly for 1 h at RT.

For three-color gSTED of endogenous mGluR5, PSD-95 and actin (Fig. S1l–n) neurons were fixed and blocked as described above. Then neurons were incubated with rabbit anti-mGluR5 (Alomone Labs) diluted 1:50 and mouse anti-PSD-95 (Neuromab) diluted 1:300 in 0.1% Triton X-100 and 5% NGS in PBS/Gly for 2 h at RT. Neurons were washed three times and incubated with anti-rabbit Atto647N (Sigma-Aldrich), anti-mouse Alexa488 (Thermo Fisher Scientific) diluted 1:250, and Phalloidin Alexa594 (Thermo Fisher Scientific) diluted 1:100 in 0.1% Triton X-100 and 5% NGS in PBS/Gly for 1 h at RT. Note that we visualized the total pool of endogenous mGluR5 using both mGluR5 antibodies.

For two-color STED of myc-mGluR5 and SEP-GluA2 (Fig. S1s–v) neurons were fixed as described above and blocked in 10% NGS in PBS/Gly for 30 min at RT. To label the surface-expressed receptors, neurons were incubated with rabbit anti-GFP (MBL) diluted 1:2000 and mouse anti-myc (Santa Cruz Biotechnology) diluted 1:500 in 5% NGS in PBS/Gly O/N at 4 °C. Neurons were washed three times and incubated with anti-rabbit Atto647N (Sigma-Aldrich) and anti-mouse Alexa594 (Thermo Fisher Scientific) diluted 1:250 in 5% NGS in PBS/Gly for 1 h at RT. All neurons were washed three times with PBS/Gly after the incubation with secondary antibodies, mounted in Mowiol mounting medium (Sigma), and imaged on the Leica SP8 microscope as described below.

## Confocal and STED microscopy

Imaging was performed with a Leica TCS SP8 STED 3× microscope using an HC PL APO ×100/NA 1.4 oil immersion STED WHITE objective.

The 488, 590, and 647 nm wavelengths of pulsed white laser (80 MHz) were used to excite Alexa488, Alexa594, and Atto647N, respectively. To obtain gSTED images, Alexa488 was depleted with the 592 nm continuous wave depletion laser, and Alexa594 and Atto647N were depleted with the 775 nm pulsed depletion laser. We used an internal Leica HyD hybrid detector (set at 100% gain) with a time gate of $0.3 \leq tg \geq 6$ ns. Images were acquired as Z stacks using the ×100 objective. Data was collected using the Leica Application Suite X (LAS-X software). Maximum intensity projections were obtained for image display and analysis. For the FKBP-rapalog-FRB heterodimerization assay DIV18-21 neurons transfected with SEP-mGluR5-FRB and 2xFKBP-Homer1c-mCherry were incubated with 1 μM rapalog diluted in extracellular imaging buffer or extracellular imaging buffer only (vehicle) for 50 min before fixation. For the quantification of mGluR5 overexpression, DIV18-21 neurons transfected with SEP-mGluR5 and stained with anti-mGluR5, confocal images were taken with a Zeiss LSM 510 with 63 × 1.40 oil objective. Images consist of a z stack of 7–9 planes at 0.39 μm interval, and maximum intensity projections were generated for analysis and display.

### Single-molecule localization microscopy using dSTORM and PALM

Neurons were fixed at DIV21 with 4% PFA/sucrose in PBS for 10 min at RT, washed three times with PBS/Gly and blocked with 10% NGS in PSB/Gly for 15 min at RT. To label the surface-expressed pool of receptors, neurons were incubated with Fluotag-X4 anti-GFP Alexa647 (Nanotag) diluted 1:250 in PBS/Gly. Neurons were washed three times in PBS/Gly and stored in PBS at 4 °C (dark) until use. Neurons were imaged in PBS containing 5 mM MEA, 5% w/v glucose, 700 μg/ml glucose oxidase, and 40 μg/ml catalase.

Dual-color SMLM data was acquired on the Nanoimager S from ONI (Oxford Nanoimaging; ONI), equipped with a ×100/NA 1.4 oil immersion objective (Olympus Plan Apo), with an effective pixel size of 117 nm, an XYZ closed-loop piezo stage, and with 405, 473, 561 and 640 nm wavelength excitation lasers. Fluorescence emission was detected using a sCMOS camera (ORCA Flash 4, Hamamatsu). Integrated filters were used to split far-red emission from blue-green-red emission, allowing simultaneous dual-color imaging. dSTORM and PALM were simultaneously performed using the 640 nm laser to bring Alexa647 to the dark state along with increasing power of the 405 nm laser to stochastically reactivate Alexa647 fluorophores and stochastically photoconvert PSD$_{FingR}$-mEos3.2 from green to red, combined with excitation of the photoconverted molecules by the 561 nm laser. Stacks of 10,000 to 20,000 images were acquired at 50 Hz with oblique illumination, which was processed using NimOS software from ONI. Before every acquisition, stacks of 30 frames were acquired with the 473 nm excitation laser to visualize SEP-mGluR5 and PSD-95FingR-mEos3.2 expression. NimOS software from ONI was used for data processing and drift correction was performed. Before each imaging session, a bead sample was used to calibrate the system and align the two channels with a channel mapping precision >8 nm. The particle tables were exported to MATLAB for analysis and images were rendered in NimOS software with 11.7 nm output pixels (sigma 1) and filtered on a minimum photocount of 300 and xy localization precision ≤30 nm for figure display.

### Single-molecule tracking with uPAINT

uPAINT for Figs. 3 and 4 was performed on the Nanoimager S from ONI (Oxford Nanoimaging; ONI), equipped with a ×100/NA 1.4 oil immersion objective (Olympus Plan Apo), an XYZ closed-loop piezo stage, and with 405, 471, 561, and 640 nm wavelength excitation lasers. Fluorescence emission was detected using a sCMOS camera (ORCA Flash 4, Hamamatsu). Stacks of 5000 frames were acquired at 50 Hz with oblique illumination. NimOS software from ONI was used for data analysis and drift correction was performed.

uPAINT for Figs. 5 and 6 was performed on a Nikon Ti microscope with a Nikon ×100/NA 1.49 Apo TIRF objective, a Perfect Focus System, a 2.5× Optovar to achieve an effective pixel size of 64 nm, and a DU-897D EMCCD camera (Andor). Imaging was performed with oblique laser illumination with a 405 nm diode laser (15 mW; Power Technology), a 491 nm DPSS laser (50 mW; Cobolt Calypso), a 561 nm DPSS laser (100 mW; Cobolt Jive), and a 640 nm diode laser (35 mW; Power Technology). Micromanager software[87] was used to control all these components. 5000 frames were acquired at 50 Hz in TIRF. Acquired image stacks were analyzed using the ImageJ plugin Detection of Molecules (DoM) v1.1.5 (https://github.com/ekatrukha/DoM_Utrecht) and drift correction was applied.

Neurons were imaged in extracellular imaging buffer containing 120 mM NaCl, 3 mM KCl, 10 mM HEPES, 2 mM CaCl$_2$, 2 mM MgCl$_2$ and 10 mM glucose, pH adjusted to 7.35 with NaOH. The GFP-booster Atto647N (Chromotek) was added before image acquisition in a concentration of 1:150.000 to 1:50.000 in extracellular imaging buffer while blocking with 0.5–1.5% BSA. Low concentrations of the GFP-booster were used to achieve temporal separation of fluorescence emission of mGluR5 molecules. Due to the low dissociation rates of the nanobody, only being limited by photobleaching, we obtained long trajectories and used a minimum track length of 30 frames (20 ms interval) for visualization and quantification. PSD masks were created from a stack of 30 frames obtained for Homer1c-mCherry using the 561 nm excitation laser.

### Live-cell spinning disk confocal imaging

The FKBP-rapalog-FRB heterodimerization assay and Ca$^{2+}$ imaging were performed on a spinning disk confocal system (CSU-X1-A1 Yokogawa; Roper Scientific) mounted on an inverted Nikon Eclipse Ti microscope (Nikon) with a Plan Apo VC ×100/1.40 NA objective (Nikon) with excitation from 491 nm Cobolt Calyspso (100 mW), 561 nm Cobolt Jive (100 mW), 642 nm Vortran Stradus (110 mW) lasers and emission filters (Chroma). The microscope is equipped with a motorized XYZ stage (ASI; MS-2000), Perfect Focus System (Nikon), and Prime BSI sCMOS camera (Photometrics), and controlled by MetaMorph software (Molecular Devices). During the image acquisition neurons were kept in extracellular imaging buffer (with or without MgCl$_2$) in a closed incubation chamber (INUBG2E-ZILCS; Tokai Hit) at 37 °C in 5% CO$_2$.

For the FKBP-rapalog-FRB heterodimerization assay neurons were transfected with SEP-mGluR5-FRB and 2xFKBP-Homer1c and imaged at DIV18–22. After a 10-min baseline acquisition, recruitment of mGluR5 to Homer1c was induced by the addition of rapalog to a final concentration of 1 μM and the SEP-mGluR5 and Homer1c-mCherry signals were imaged every 5 min for another 40 min. Multiple Z stacks (seven planes) were obtained, with 0.5 μm intervals to acquire 3 μm image stacks.

For the heterodimerization assay combined with Ca$^{2+}$ imaging, neurons were transfected with SNAP-mGluR5-FRB, 2xFKBP-Homer1c-mCherry, and GCaMP6f or SNAP-mGluR5, 2xFKBP-Homer1c-mCherry and GCaMP6f for control neurons. At DIV21-22, before each acquisition a coverslip with neurons was labeled with a SNAP JF646 cell-impermeable dye (JF646i; Janelia/Tocris) diluted 1:2000 in supplemented medium for 30 min followed by a single wash with extracellular buffer. Neurons were transferred to the imaging chamber containing extracellular imaging buffer without MgCl$_2$ and with 3 μM tetradotoxin citrate (TTX; Tocris) to block action potentials and relieve the NMDA receptor pore block. At the start, Z stacks (7 planes) were obtained of the SNAP-mGluR5-FRB JF646i and 2xFKBP-Homer1c-mCherry channels, with 0.5 μm intervals to acquire 3 μm image stacks. This was shortly followed by a 50-second stream of the GCaMP6f signal, acquired at 50 ms intervals (20 Hz) (referred to as "before" rapalog). Then, rapalog was added to the imaging chamber to a final concentration of 1 μM and incubated for 30 min, the time we established is required for the synaptic recruitment of mGluR5 to

reach a plateau. Again, this was followed by imaging stacks of SNAP-mGluR5 JF646i and Homer1c-mCherry and a stream of GCaMP6f (referred to as after rapalog). Maximum intensity projections were obtained of the mGluR5 and Homer1c stacks for image display and analysis.

For Ca$^{2+}$ imaging experiments (without FKBP/FRB recruitment), neurons were transfected with GCaMP6f and mCherry as a fill marker and imaged at DIV21-22. Neurons were transferred to the imaging chamber containing extracellular imaging buffer without MgCl$_2$ and with 3 μM TTX (Tocris). A Z stack was obtained of mCherry and a 50-second baseline stream of GCaMP6f (same imaging settings as described above). In the drug treatment experiment, the baseline acquisition was followed by the application of 100 μM (S)−3,5-dihydroxyphenylglycine (DHPG; Tocris) for 5 min, a wash-out, 5 min recovery, and an additional 50-second stream of GCaMP6f. Then, this was followed by a 5-min incubation of 50 μM DL-2-Amino-5-phosphonopentanoic acid (DL-AP5; Tocris) and another 50-second stream of GCaMP6f.

## Quantification and statistical analysis

**Quantification of spine enrichment and mGluR5 expression.** To assess the spine enrichment of surface mGluR5 and mGluR5/STGtail variants, the Atto647N intensity from confocal images were quantified as mean spine intensity divided by mean dendritic shaft intensity. For each neuron, circular regions of interest (ROIs) were traced on multiple dendritic spines to measure spine intensity and for each selected spine an ROI in the dendrite at the base of the spine was measured as dendritic shaft intensity. Background intensity was subtracted. For Figs. 1 and S7 the spine enrichment of mCherry and Homer1c-mCherry was determined using the same spine and dendrite ROIs as used for mGluR5. To analyze mGluR5 overexpression levels, the average intensity of mGluR5 staining was measured in dendritic segments of SEP-mGluR5 expressing neurons and neighboring untransfected neurons in the same field-of-view.

**STED imaging analysis.** To assess the localization of mGluR5 relative to Homer1c, PSD-95, GluA2, or Phalloidin, line profiles along spines were drawn using ImageJ software. To quantify the localization of mGluR5, mGluR5ΔC, mGluR5WT-STGtail, and mGluR5ΔC-STGtail in spines, all images were scrambled and blinded. Per neuron, a minimum of 20 spines were selected based on the Homer1c channel and the localization of mGluR5 was determined using the merged image of confocal-resolved Homer1c and gSTED-resolved mGluR5 in ImageJ software. The localization of mGluR5 could be categorized as spines with (1) synaptic enrichments, (2) synaptic and perisynaptic enrichments, (3) perisynaptic enrichments or (4) a homogeneous distribution of mGluR5. Per category, the percentage of spines was plotted and the statistical significance was determined within each category and all conditions were compared to mGluR5WT.

**Single-molecule localization analysis.** The maximum projections of the 30 frames acquired in the green channel were used to select all spines and save these as separate ROIs using ImageJ software. The molecules from the ROIs were extracted and used for further analysis and were filtered on a localization precision <20 nm. Furthermore, molecules that were in the fluorescent state longer than 1 frame were filtered out by tracking with a radius of 58.5 nm (0.5 pixels). PSD$_{FingR}$ clusters were identified using DBScan[27] executed in MATLAB. PSD$_{FingR}$ clusters with a density >1200 molecules per μm (epsilon 0.35 and >50 localizations) were used for further analysis and the PSD border was defined using the alpha shape. The distance of individual localizations to the nearest PSD border (up to 1 μm distance) were computed and plotted as a frequency distribution. Rings were calculated as a fraction of the PSD border polyshape (is 1) defined by DBScan with two rings inside the PSD: 0–0.5 and 0.5–1 and six rings outside the PSD, with

three rings approximating the perisynaptic zone: 1–1.5, 1.5–2, and 2–2.5 and three rings defining the extrasynaptic region: 2.5–3, 3–3.5 and 3.5–4. Per ring, the number of mGluR5 localizations was determined and the fraction of mGluR5 localizations per ring was calculated. To correct for the different sizes of ring 1 to 8, we further calculated the fraction of the area covered by each ring. The fraction of mGluR5 localizations was divided by the fraction of ring area, and normalized to 1. Then we also assessed the existence of mGluR5 clusters using DBScan and a density of >480 molecules per μm (epsilon 0.35 and >20 localizations). The border-to-centroid distance from PSD to mGluR5 cluster was calculated and plotted as a frequency distribution. To detect clusters in spines using SR-Tesseler we set the criteria to >20 localizations and area >400 nm$^2$ (localization precision is 20 nm) and used a density factor of 2. To detect all objects in spines and dendrites we only applied a density factor of 2 and removed all other thresholds.

**Single-molecule tracking analysis.** Using MATLAB, molecules with a localization precision <50 nm were selected for analysis and background localizations were removed by outlining the neuron based on the obtained SEP-mGluR5 widefield image. Tracking was achieved using custom algorithms in MATLAB described previously[88]. For tracks consisting of ≥4 frames the instantaneous diffusion coefficient was estimated. The first three points of the MSD with the addition of the value 0 at MSD(0) were used to fit the slope using a linear fit. Tracks with a negative slope were not used for further analysis. The diffusion coefficient was estimated based on the fit using:

$$MSD = 4D\Delta t \tag{1}$$

Only tracks of at least 30 frames were selected for further analysis. Tracks were classified as immobile when the ratio between the radius of gyration and mean step size was smaller than 2.11[31]. This ratio was calculated using:

$$\text{ratio} = \frac{\sqrt{\pi/2} \cdot \text{radius of gyration}}{\text{mean stepsize}} \tag{2}$$

The PSD mask was created based on the maximum intensity projection of Homer1c-mCherry. Peaks in intensity were detected after which a FWHM-like boundary was defined for each PSD. An expanded PSD mask of 200 nm around the PSD mask was created to define the perisynaptic zone. Tracks were assigned to the synaptic group if ≥80% of the localizations of the track overlapped with the PSD. Perisynaptic tracks had to overlap ≥60% with the perisynaptic zone and <80% with the PSD, and transient perisynaptic tracks overlapped >0% and <60% with the perisynaptic zone. Entries and exits per perisynaptic trajectory were derived based on their overlap with the PSD mask. The perisynaptic tracks were categorized into three groups: captured, returned or escaped. The 'captured' tracks were the tracks that started within the (peri)synaptic region or entered this region but never left. The 'returned' tracks were at least once outside the (peri)synaptic region over the course of the track but ended up within the (peri)synaptic region. Lastly the 'escaped' group contains the tracks that crossed the perisynaptic region, but ended outside.

Transient confinement analysis on mobile trajectories was done in MATAB using slightly modified scripts from a previously published MATLAB implementation[89] based on the algorithm reported by[33,90]. Briefly, transient confinement was detected in a trajectory based on the probability (ψ) of a molecule staying within a region of radius (R) for a period of time (t):

$$\log(\psi) = 0.2048 - 2.5117Dt/R^2 \tag{3}$$

where D is the maximum of the instantaneous diffusion coefficients estimated for each sub-trajectory of Δ10. This probability was translated into a confinement index L, the larger the value of L, the

greater the probability that the observed part of the trajectory is not of Brownian origin. The regions where the confinement index is above the critical $L$ for critical time $Tc$ are identified as confinement zones. Parameters used in the analysis are: $Lc = 4$, $Sm = 15$, $\alpha = 0.5$, $Tc = 0.2$ s (10 frames). The confinement zones are further analyzed for size and duration of confinement and diffusion coefficient in and outside confinement zones.

Confinement maps were created based on the detected confinement radius for each confinement zone. Each confinement zone was stored as a 2D Gaussian with the radius as FWHM. The final matrix was plotted with a color code, where higher values indicate confinement hotspots because there are multiple Gaussians on top of each other. For the immobile tracks the center of the track coordinates was determined and a 2D Gaussian with a fixed FWHM of 75 nm was plotted. The distance between a confinement zone and a PSD was defined as the shortest distance between the center of a confinement zone or immobile track to the nearest PSD border.

**mGluR5-FRB to FKBP-Homer recruitment analysis.** The maximum intensity projections were corrected for XY drift over time using the ImageJ plugin "StackReg." We quantified the SEP-mGluR5-FRB recruitment to 2xFKBP-Homer1c-mCherry over a time-period of 40 min, after a 10-min baseline period. The Homer1c timelapses were used to select ROIs using ImageJ. First, image noise was reduced by applying a gaussian blur with sigma = 1 and background subtraction with a rolling ball of 50. Subsequently, the time-lapse images were subjected to thresholding based on the $t = -10$-min image to isolate all PSDs. Then, a mask followed by a selection of all PSDs was created for each timepoint and saved as ROIs. The raw time-lapse images of Homer1c and mGluR5 were used to measure the signal intensity within the ROIs at the different time points. To obtain the change in relative fluorescence intensity ($\Delta F/F_0$) over time, the intensity relative to $t = -10$ min was calculated and for visualization all values were subtracted by 1. The increase in mGluR5 intensity upon rapalog application, measured within PSDs marked by Homer1c, was best explained by a one-phase association function, fitted using Graphpad Prism.

**Ca2+ imaging analysis.** To analyze the $Ca^{2+}$ imaging data, a circular ROI was drawn around every spine within the field of view, clearly separated from the dendritic base and in focus, regardless of activity levels. Using ImageJ software, the mean intensity value within each ROI was measured for all 1000 frames (50 ms streams). Then, this data was analyzed using custom MATLAB scripts based on[46]. Peaks of mSCTs were detected and measured if the 2-point slope was greater than $meanslope + 2 \cdot STDslope$, and amplitude greater than 0.035 $\Delta F/F_0$. Using these and several other criteria described in more detail in Reese and Kavalali (2015), peaks were consistently detected, disregarding background noise or single high point artifacts. Then for each spine the mSCT frequency and for each peak the mSCT amplitude ($\Delta F/F_0$) was calculated. To measure average decay times, all $\Delta F/F_0$ values of detected peaks were loaded into Clampfit 10.3 (Molecular Devices) and average mSCT traces were made for each neuron by aligning all peaks before and all peak after rapalog application. Next, a single-exponential fit line was obtained from the decay phase of the average mSCT traces. The single-exponential fit lines were plotted in Graphpad Prism and the decay times (tau), the time in seconds required to decay to $(\frac{1}{e})\Delta F$, were calculated. For measurements of mSCT frequency, tau, and amplitude spines with at least one mSCT were included and data is presented as the mean mSCT frequency (Hz) per neuron, mean mSCT tau (s) per neuron, and mean mSCT amplitude ($\Delta F/F_0$) per neuron, respectively. While in the frequency analysis correcting for the variability in the number of ROIs (spines) in the field of view, all spines (also without activity) were included and data are presented as the mean mSCT frequency (Hz) per spine. To test the statistical

significance of the change in decay times and amplitude upon rapalog application, only neurons that had at least one mSCT both before and after rapalog application could be used.

**Statistical analysis.** All statistical tests used and significance in this study are described in the main text and figure legends. In the figures, * indicates significance based on the condition effect and when comparing more than two groups, * indicates significance based on the multiple comparison test. In all figures * was used to indicate a $P$ value <0.05, ** for $p < 0.01$, and *** for $p < 0.001$. If normally distributed, data are represented as mean ± SEM. Non-normal data are represented as median with 95% confidence interval (CI), with the [lower upper] limits of the 95% CI mentioned in the main text. Each experiment was replicated in cultures from at least three independent preparations of hippocampal neurons. The $n$ indicated in the figure legends are the number of neurons used for analysis, unless stated otherwise. Statistical analysis and graphs were prepared in GraphPad Prism and figures were generated in Adobe Illustrator CC.

### Reporting summary
Further information on research design is available in the Nature Portfolio Reporting Summary linked to this article.

## Data availability
The data that support the findings of this study are available from the authors upon request. Source data are provided with this paper.

## Code availability
The MATLAB code used in this study are based on previously published MATLAB algorithms as referenced accordingly. The custom MATLAB scripts are available from the authors upon request.

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

## Acknowledgements

We would like to thank all members of the MacGillavry lab and Eline Penners for helpful discussions and Arthur de Jong, Yolanda Gutierrez, and Lisa A.E. Catsburg for critical reading of the manuscript. This work was supported by the Netherlands Organization for Scientific Research (the Graduate Program of Quantitative Biology and Computational Life Sciences) to N.S. and the European Research Council (ERC-StG 716011) to H.D.M.

## Author contributions

Conceptualization, methodology, N.S. and H.D.M.; validation & formal analysis, N.S. and M.W.; investigation, N.S.; resources, H.D.M.; writing—original draft, N.S.; writing—review, H.D.M. and M.W.; visualization, N.S.; supervision, H.D.M.; funding acquisition, H.D.M. and N.S.

## Competing interests

The authors declare no competing interests.
