## [Peer Review File · Nature Communications]

mGluR5 is transiently confined in perisynaptic nanodomains to shape synaptic functionREVIEWER COMMENTS

Reviewer #1 (Remarks to the Author):

In this work, using state of the art microscopy technologies, Scheefhals and collaborators re-investigated the subsynaptic organization of mGluR5 and its dynamics at excitatory synapses. Their experiments confirm a largely perisynaptic expression of the mGluR5 receptor, revealing that 55 % of the mGluR5 population is organized as clusters juxtaposed to the PSD. A small fraction of the mGluR5 population (15%) is statically localized in the PSD and 30% is highly mobile, transiting between the extra- and perisynaptic compartments. This part of the study, including experimental design, implementation of the experiments, analysis and interpretation are carried-out with great rigor. The message, while not fundamentally new, adds much to the field by providing a more accurate picture of receptor organization at the synapse.

-> Comment 1: This heterogeneous localization suggests different roles for mGluR5: within the PSD, in a peri- or extra-synaptic position, which need to be discussed.

In a second part, the authors try to understand which molecular mechanisms could be responsible for this confinement of the receptor at the perisynaptic level. By making chimeric proteins they demonstrate 1) that the C-term-deleted mGluR5 (mGluR5DC) has a much more diffuse expression than mGluR5 and 2) that a synaptic addressing tag, STGtail, is sufficient to address the mGluR5DC to the PSD but has no effect on the full length mGluR5.

-> Comment 2: The first results suggest that the C-terminal part of the receptor is necessary for a perisynaptic confinement of mGluR5. The second set of results favors the hypothesis that the C-term may support a synaptic exclusion mechanism. As it stands, this conclusion is a bit premature as it could simply be that the synaptic addressing tag is not properly exposed in the mGluR5STGtail chimera. However, the fact that mGluR5 Ctail is sufficient for synaptic exclusion is a very interesting hypothesis which should be directly tested by fusing the Ctail of mGluR5 to a single transmembrane domain and assessing if this construct is confined to perisynaptic sites. This hypothesis, if verified, would suggest that there is a molecular mechanism centred on the Ctail of the receptor that allows its confinement to the perisynaptic level, presumably as discussed by the authors an unknown protein partner that constrains receptor diffusion.

In the last part of the work, using an heterodimerization system allowing robust and rapid recruitment of mGluR5 to the synapse the authors studied the effect of mGluR5 positioning on synaptic signaling. They show that dimerization of SEP-mGluR5-FRB with FKBP-Homer1c increases the enrichment of mGluR5 in spines, and significantly increased mSCT peak intensities and frequency.

-> Comment 3: It would be interesting to know whether this synaptic increase only lies on the accumulation of mGluR5 in the PSD or if increasing homer1c-mGluR5 interaction enhanced mGluR5 signaling. To test this hypothesis, first, the increase in mSCT can be normalized by mGluR5 expression at the PSD; an second, the experimental heterodimerization to drag mGluR5 in the PSD could be performed using another synaptic protein, like PSD95-FKBP.

Finally, a large part of the discussion is dedicated to the mechanisms that could explain mGluR5 perisynaptic confinement. As the authors themselves state, the C term of the receptor could play a role but this does not exclude the involvement of other domains of the receptor in its perisynaptic expression. Regarding the proteins involved in this specific targeting, they are to date totally unknown. However, a part of the discussion is dedicated to the exclusion of the role of Homer in this process.

-> Comment 4: To claim that Homer is not involved in this process is quite unexpected, to say the least, because it is inconsistent with numerous studies in the field and the results presented here

could even go against this assertion. Indeed, one can note a partial overlap of Homer1c with GluA2 (Fig S1 L-O) but also with mGluR5 (Fig. 1 D-H) in contrast to mGlu5 and PSD95 that do not co-localize at all (Fig. 1 J-M). In other words, the expression of Homer1c seems to extend to the synaptic periphery (even if it is not as strong, I agree). At this position (synaptic periphery) Homer could just as well as other proteins play a role of anchoring mGluR5, role that it would not have at the synaptic level because it would interact with other partners? These are possibilities that cannot be ruled out and should be discussed as well, or this part of the discussion - out of the scope of the present study - should be deleted.

In conclusion, it is a serious and rigorous work, of great interest, which deserves to be published in nature communication providing substantial revisions.

Reviewer #2 (Remarks to the Author):

Previous studies have found mGluR5 receptors organized in perisynaptic nano-domains with respect to the postsynapse. Here, Scheefhals et al. confirm and extend on these findings to disclose the dynamic organization of mGluR5 at synapses, showing that mGluR5 movement is perisynaptically confined, with the location and dynamics controlled through the c-terminal domain. The authors move on to investigate the functional implications of this for synaptic calcium signaling measured in dendritic spines, which in my view is the most interesting aspect and where the key novelty for the field lies. The organization of synaptic receptors and channels into nano-domains and the functional roles of these is a timely and interesting subject of relevance for a broad audience in the field of neuroscience. The manuscript undoubtedly reflects a huge work effort and extensive experience with state-of the art microscopy and labeling approaches. It is very clearly written, and the figures illustrative. The presented experiments appear well performed and well described. As far as I can judge the manuscript adequately sites relevant literature in the field. My main concern for endorsing the manuscript is the design of the key experiments to investigate the functional role of mGluR5 in regulating ionotropic synaptic signaling, which I am not sure represents a physiological scenario.

Main critique

While I find the mGluR5 nano-domain organization, perisynaptic confinement, and receptor dynamics data convincing, as well as the role of the c-terminal domain in orchestrating this, I am not convinced about the functional implications for ionotropic signaling based on the calcium imaging experiments performed, or that mGluR5 is important for shaping synaptic ionotropic signaling. To me, it is not shown that “mGluR5 is transiently confined in perisynaptic nanodomains to shape synaptic transmission”, as the title states. Or the corresponding discussion sentence “we present mechanistic understanding of how the CTD of mGluR5 controls its dynamic organization in perisynaptic nanodomains, as well as preventing mGluR5 from entering the synapse, allowing mGluR5 to finely tune synaptic signaling.”, and similar sentences about effects on synaptic signaling around line 369. My concern is that the experiments that disclose an augmentation of synaptic calcium signaling rely on overexpressing (FRG-tagged) mGluR5 and inducing “synaptic recruitment” by forcing tagged mGluR5 into the PSD (through FKBP-tagged Homer1c and the rapalog approach) on a background of endogenous mGluR5 receptors that (presumably) remain perisynaptic. It is not properly justified that these conditions (overexpression and forced re-location to the PSD) represent a plausible physiological scenario and that a corresponding role of mGluR5 can be expected in the living brain. Indeed the authors actually show that endogenous mGluR5 is very unlikely to undergo “synaptic recruitment” and enter the PSD, so it is unclear to me what the rapalog experiment tells us about the putative physiological role of synapse-related mGluR5 receptor nano-domains in shaping synaptic signaling.

To address this, I suggest the authors seek to corroborate their data by showing that endogenous mGluR5 receptors can have an impact on synaptic calcium signaling (or electrophysiologically measured synaptic signals).

Also, it would be relevant to repeat the current calcium imaging and rapalog experiment while

blocking mGluR5 to test whether the altered calcium signaling is indeed a direct effect of mGluR5 and not an indirect effect i.e. from crowding in the PSD.

Other comments

- Relating to my main comment, in Fig 7 only overexpressed mGluR5 are illustrated in the cartoon graphic, is there not still also endogenous mGluR5? The ratio of endogenous to overexpressed mGluR5 receptors is critical for interpreting these experiments and should be somehow quantified, even if by simple immunocytochemistry.
- The uPAINT experiments were also performed after overexpression of (SEP-) mGluR5, again without knowing the overexpression level. Could the uPAINT experiments not be performed targeting endogenous mGluR5 using a fluorophore-tagged mGluR5 antibody, or is this not possible?
- Given the uncertainty of the source of the observed calcium transients, it would be informative to at least confirm they were synaptic NMDAR dependent by investigating if they are blocked by NMDAR blockers/antagonists (or alternatively by including a control group with physiological concentration of Mg in the buffer).
- Line 245: "Although not significant, synaptic mGluR5 Δ C-STGtail diffused at a slower rate compared to mGluR5WT, suggesting that the addition of the STGtail increased the retention in the PSD...". If the difference is not significant then the authors cannot say that it diffused slower and should modify the text accordingly to be more true to the observed result.
- Throughout, I suggest showing SD in error bars as it holds valuable information about variation (physiological and methodological), rather than the less informative SEM.
- In Fig. 1C, perhaps specify that the "ratio" is the "intensity ratio"?
- Any possible non-normal data should be depicted with median instead of mean, and not with SEM. For example, are data depicted in Fig. S8D normal?
- Line 77, 78, 79 and throughout: Instead of using simply "anti-PSD-95", "anti-mGluR5", and so forth, please spell out whether this refers to an antibody or some other label.
- Line 303: "form" should be "from".
- Line 666: "...room RT,..."
- Line 807 and throughout text figures and suppl figures: To refer to line profile intensities in 2D images "line profile" or similar is better than the currently used "linescan" or "line scan", which are more often used to refer to dynamic scanning of a line for image acquisition.

Reviewer #3 (Remarks to the Author):

Scheefhals et al. aims to determine the behavior of the mGluR5 at the synapse by using super-resolution techniques. The main conclusions are that mGluR5 have some transient trapping at the perisynapse but is mainly excluded from the PSD, the Cterm tail of mGluR5 is implicated in the trapping, the overaccumulation of mGluR5 at the PSD would increase the frequency of calcium entry at spine.

The paper uses a wide range of techniques and way of labeling the receptor. The images and the figures are of a high quality. The overall story is very elegant and the conclusion of such studies is strongly awaited by the neuroscience community. Indeed, the dynamic nanoscale organization of both AMPAR and NMDAR has been achieved, but that of mGluR is still lacking. However, some significant experiments/ analyses should be performed to validate the hypothesis.

Major points:

- Differences between clusters and single particles. With super-resolution, couple of detections close each other can be either a cluster of proteins (as for AMPAR or NMDAR) or the multiple detections due to the blink effect of the molecule or the various frames of a single and immobile track. Moreover, it is now regularly accepted that mGluR are mainly dimeric. It is crucial here to demonstrate whether the objects correspond to a cluster of detections or to a cluster of proteins. The ultimate question is to determine whether there is some transient trapping of a dimer or some well define nano-organization at the perisynapse.

This point is crucial for concluding between a clustered organization or a homogeneous organization

as argued in Goncalves et al. 2020 in PNAS.

- Around the same idea, the distribution of the size of the individual mGluR5 is not described, the pointing accuracy of a single detection and the resolution of a single particle differs. In the figures, in STED and STORM, many 80 nm clusters resemble to single molecules. It might be worth comparing the synaptic “cluster/ single molecule” with that on dendrite.

- The ring analysis (figure 2E) is a bit confusing to me, if the PSD is on 2 rings (around 200 nm), it is surrounded by the two perisynaptic ring (200 nm from each side), it makes 600 nm which is the size of the spine. Thus, the other rings tend to exhibit a lower density of mGluR because it is out of the neuron (except at the neck localization).

- The vast majority of the experiments are performed by overexpressing the mGluR5 which can affect the density and distribution, but more importantly, by putting a GFP (even in the elegant knock in experiments) which have a quite large size. If we had the size of the GFP to the size of the extracellular part of the mGluR, is it possible that a form of exclusion is created at the PSD level which is already full of adhesion proteins and where the presynapse is probably the closest from the postsynapse?

On the same line, if the exclusion of the PSD (which has been already shown in various paper), is combined to the synaptic enrichment due to local endo/ exocytosis, and the shape of the neck which limit the exchange between spine and dendrite, it is possible that local enrichment at the perisynapse is only a passive result and not a specific trapping. This does not change the quality of the experiments but more their interpretations.

- The u-PAINT experiments are well done and important, the transient trapping of some receptors is clearly demonstrated. However, there is some bias in the analysis. (1) It is normal that the transient perisynaptic move more because they have been chosen on this property, they are not trapped at the perisynapse, so they are more mobile. It could be interesting to have the distribution of the dendritic receptors, and more particularly on the knock in ones, to avoid artifact from overexpression. Are there trapped receptors on the dendrites? (2) The MSD curve of the figure 3H (corresponding to the legend 3I) start at the 0 value which is not normal for a MSD which should start from a positive value corresponding to the pointing accuracy. And the curves seem to be a fit of the data, what is use for this fit?

- Concerning the role of the C-term of mGluR, I wonder whether the correct control might not be the delta-C-term fused to the STG tail, because, of course the fusion of the entire tail of stargazin to the already long tail of mGluR could have some exclusion effects at the PSD because of the size. Another possibility could be to add a PSD binding site at the end of the mGluR tail.

- Finally, the effect of the recruitment of mGluR at the PSD. These experiments are difficult and clever. The total amount of mGluR per spine need to be done. For now, it is not known if the increase of calcium is due to the larger amount of mGluR at the spine or to its re-localization at the PSD.

Finally, it is difficult to understand why the frequency is affected

REVIEWER COMMENTS

Reviewer #1

In this work, using state of the art microscopy technologies, Scheefhals and collaborators re-investigated the subsynaptic organization of mGluR5 and its dynamics at excitatory synapses. Their experiments confirm a largely perisynaptic expression of the mGluR5 receptor, revealing that 55 % of the mGluR5 population is organized as clusters juxtaposed to the PSD. A small fraction of the mGluR5 population (15%) is statically localized in the PSD and 30% is highly mobile, transiting between the extra- and perisynaptic compartments. This part of the study, including experimental design, implementation of the experiments, analysis and interpretation are carried-out with great rigor. The message, while not fundamentally new, adds much to the field by providing a more accurate picture of receptor organization at the synapse.

-> Comment 1: This heterogeneous localization suggests different roles for mGluR5: within the PSD, in a peri- or extra-synaptic position, which need to be discussed.

We thank the Reviewer for the careful assessment of the manuscript and the positive comments acknowledging the strength of the data and the value of this study for the field. We addressed the Reviewer's comment 1 and expanded our discussion on this matter (see lines 395-400).

In a second part, the authors try to understand which molecular mechanisms could be responsible for this confinement of the receptor at the perisynaptic level. By making chimeric proteins they demonstrate 1) that the Cterm-deleted mGluR5 (mGluR5DC) has a much more diffuse expression than mGluR5 and 2) that a synaptic addressing tag, STGtail, is sufficient to address the mGluR5DC to the PSD but has no effect on the full length mGlu5R.

-> Comment 2: The first results suggest that the Cterminal part of the receptor is necessary for a perisynaptic confinement of mGluR5. The second set of results favors the hypothesis that the C-term may support a synaptic exclusion mechanism. As it stands, this conclusion is a bit premature as it could simply be that the synaptic addressing tag is not properly exposed in the mGluR5STGtail chimera.

This is an interesting point, also raised by Reviewer #3. Indeed, the location of the synaptic targeting motif in the STG tail sequence might not be exposed properly in the mGluR5STGtail chimera. We therefore tested whether the location of the STG tail in the chimeric construct affects its ability to target mGluR5 to the synapse. To do so, we designed a mGluR5-STGtail_{split} construct where the STGtail sequence was inserted upstream of the mGluR5 CTD sequence, while the STG PDZ ligand was left at the extreme C-terminal position to retain its ability to bind synaptic scaffolds. Importantly, we found that this mGluR5-STGtail_{split} chimera was similarly excluded from the PSD. The spine enrichment of this construct was in fact not significantly different from the original mGluR5-STGtail chimera (Figure S6f-h, I in the revised manuscript). We describe these experiments in the revised manuscript at line 252-270.

However, the fact that mGlu5R Ctail is sufficient for synaptic exclusion is a very interesting hypothesis which should be directly tested by fusing the Ctail of mGluR5 to a single transmembrane domain and assessing if this construct is confined to perisynaptic sites. This hypothesis, if verified, would suggest that there is a molecular mechanism centred on the Ctail of the receptor that allows its confinement to the perisynaptic level, presumably as discussed by the authors an unknown protein partner that constrains receptor diffusion.

We indeed predict that the mGluR5 CTD is sufficient to exclude membrane proteins from the synapse. To strengthen this hypothesis, we performed a number of additional experiments. Following the Reviewers' suggestion, we fused the C-terminal domain (CTD) of mGluR5 to a single transmembrane domain (SEP-TM-mGluR5CTD). However, we found that this construct is not properly expressed in neurons (see figure below), probably due to aberrant trafficking and aggregation of this construct, preventing further analysis.

As an alternative approach, we assessed whether the mGluR5 CTD is sufficient to counteract the synaptic targeting effect of the STGtail. We therefore designed a pDisp-CTD-STGtail_{split} chimera construct containing both the mGluR5 CTD and the STGtail (Figure S6i). This construct was properly expressed and was indeed significantly less enriched in spines compared to the pDisp-STGtail construct (Figure S6i-l). In fact, the enrichment of pDisp-CTD-STGtail_{split} was not significantly different from mGluR5-STGtail and mGluR5-STGtail_{split}, indicating that the mGluR5 CTD is sufficient to exclude membrane proteins from the synapse. However, we did observe that in a subset of neurons the spine enrichment was similar to mGluR5 Δ C-STGtail. This suggests that other domains of mGluR5 might in addition contribute to the full synaptic exclusion observed for the full-length mGluR5, possibly by steric hindrance or molecular crowding mechanisms via the N-terminal and transmembrane domains. We describe these experiments in the revised manuscript at line 252 –270.

In the last part of the work, using an heterodimerization system allowing robust and rapid recruitment of mGluR5 to the synapse the authors studied the effect of mGluR5 positioning on synaptic signaling. They show that dimerization of SEP-mGluR5-FRB

with FKBP-Homer1c increases the enrichment of mGluR5 in spines, and significantly increased mSCT peak intensities and frequency.

-> Comment 3: It would be interesting to know whether this synaptic increase only lies on the accumulation of mGluR5 in the PSD or if increasing homer1c-mGluR5 interaction enhanced mGluR5 signaling. To test this hypothesis, first, the increase in mSCT can be normalized by mGluR5 expression at the PSD; an second, the experimental heterodimerization to drag mGluR5 in the PSD could be performed using another synaptic protein, like PSD95-FKBP.

We agree that this is an interesting point to address. We initially set out to indeed correlate the level of recruitment to the effects on mSCaT frequency and intensity. Nevertheless, we found that this was technically not achievable, mostly because in these time-lapse experiments combining three channels (GCaMP, Homer1c and mGluR5) the mGluR5 signal was bleached significantly, preventing us from accurately quantifying the relative expression levels of mGluR5 at the PSD before and after addition of rapalog. The second experimental approach we think is interesting but would introduce additional parameters to consider. For instance, PSD95 overexpression is expected to severely affect synaptic calcium signaling. Additionally, using other synaptic anchors to recruit mGluR5 would indeed be a means to confirm our findings. But, albeit less directly, the synaptic recruitment of mGluR5 would nonetheless still confer an increase in the association of mGluR5 with Homer1c, and we think such an experiment would not decisively exclude that indeed an increased association of mGluR5 with Homer1c increases the signaling capacity of mGluR5. More broadly speaking, we agree that the increase in mGluR5-mediated signaling after synaptic recruitment cannot simply be explained by a single effector. This effect could indeed be mediated via Homer1c or via direct actions of mGluR5 on the NMDA receptor. While decisive experiments to address this in more detail go beyond the scope of the present study, we did perform additional experiments demonstrating that endogenous mGluR5 can potentiate signaling through the NMDA receptor (also see our response to Reviewer #2 and experimental data in Figure S8d-f).

Finally, a large part of the discussion is dedicated to the mechanisms that could explain mGluR5 perisynaptic confinement. As the authors themselves state, the C term of the receptor could play a role but this does not exclude the involvement of other domains of the receptor in its perisynaptic expression. Regarding the proteins involved in this specific targeting, they are to date totally unknown. However, a part of the discussion is dedicated to the exclusion of the role of Homer in this process.

-> Comment 4: To claim that Homer is not involved in this process is quite unexpected, to say the least, because it is inconsistent with numerous studies in the field and the results presented here could even go against this assertion. Indeed, one can note a partial overlap of Homer1c with GluA2 (Fig S1 L-O) but also with mGluR5 (Fig. 1 D-H) in contrast to mGlu5 and PSD95 that do not co-localize at all (Fig. 1 J-M). In other words, the expression of Homer1c seems to extend to the synaptic periphery (even if it is not as strong, I agree). At this position (synaptic periphery) Homer could just as well as other proteins play a role of anchoring mGluR5, role that it would not have at

the synaptic level because it would interact with other partners? These are possibilities that cannot be ruled out and should be discussed as well, or this part of the discussion - out of the scope of the present study - should be deleted.

We did not intend to claim that the interactions between Homer1c and mGluR5 are irrelevant. Indeed, perhaps the peripheral population of Homer1c molecules interacts with mGluR5 to anchor the receptor close to the PSD. We adjusted and shortened the discussion on this aspect. See lines 363-367

In conclusion, it is a serious and rigorous work, of great interest, which deserves to be published in nature communication providing substantial revisions.

Reviewer #2

Previous studies have found mGluR5 receptors organized in perisynaptic nano-domains with respect to the postsynapse. Here, Scheefhals et al. confirm and extend on these findings to disclose the dynamic organization of mGluR5 at synapses, showing that mGluR5 movement is perisynaptically confined, with the location and dynamics controlled through the c-terminal domain. The authors move on to investigate the functional implications of this for synaptic calcium signaling measured in dendritic spines, which in my view is the most interesting aspect and where the key novelty for the field lies.

The organization of synaptic receptors and channels into nano-domains and the functional roles of these is a timely and interesting subject of relevance for a broad audience in the field of neuroscience. The manuscript undoubtedly reflects a huge work effort and extensive experience with state-of the art microscopy and labeling approaches. It is very clearly written, and the figures illustrative. The presented experiments appear well performed and well described. As far as I can judge the manuscript adequately sites relevant literature in the field.

We thank the reviewer for the thorough evaluation of our manuscript and constructive discussion. We have tried to address all the Reviewer's comments in the revised manuscript

My main concern for endorsing the manuscript is the design of the key experiments to investigate the functional role of mGluR5 in regulating ionotropic synaptic signaling, which I am not sure represents a physiological scenario.

Main critique

While I find the mGluR5 nano-domain organization, perisynaptic confinement, and receptor dynamics data convincing, as well as the role of the c-terminal domain in orchestrating this, I am not convinced about the functional implications for ionotropic signaling based on the calcium imaging experiments performed, or that mGluR5 is important for shaping synaptic ionotropic signaling. To me, it is not shown that "mGluR5 is transiently confined in perisynaptic nanodomains to shape synaptic transmission",

as the title states. Or the corresponding discussion sentence “we present mechanistic understanding of how the CTD of mGluR5 controls its dynamic organization in perisynaptic nanodomains, as well as preventing mGluR5 from entering the synapse, allowing mGluR5 to finely tune synaptic signaling.”, and similar sentences about effects on synaptic signaling around line 369.

We did not intend to overstate our claims and regret if that led to misunderstanding of our conclusions. We adjusted the title and statements in the main text, performed additional experiments and expanded our explanation of how we meant to interpret our results to support our ideas of the functional implications of the particular nanoscale mGluR5 distribution for synaptic signaling.

My concern is that the experiments that disclose an augmentation of synaptic calcium signaling rely on overexpressing (FRG-tagged) mGluR5 and inducing “synaptic recruitment” by forcing tagged mGluR5 into the PSD (through FKBP-tagged Homer1c and the rapalog approach) on a background of endogenous mGluR5 receptors that (presumably) remain perisynaptic.

The untagged, endogenous mGluR5 pool indeed presumably remains unaffected after the addition of rapalog, although these receptors might dimerize with overexpressed mGluR5 and could in principle also be recruited to the PSD. Even more so, Figure 7 shows that even though mGluR5 is largely recruited to the synapse upon the addition of rapalog, overexpressed mGluR5 is also not completely removed from the perisynaptic region. However, the aim of the heterodimerization experiment in Figure 8 is to test whether relocalization of mGluR5 to the synapse, increasing the number of receptors closer to the release site of glutamate, leads to aberrant calcium signaling during spontaneous release events. Therefore, for the interpretation of the results in Figure 8 the amount of mGluR5 remaining in the perisynaptic region, being overexpressed mGluR5 that is not or inefficiently recruited to the PSD or being endogenous mGluR5, is not relevant.

It is not properly justified that these conditions (overexpression and forced re-location to the PSD) represent a plausible physiological scenario and that a corresponding role of mGluR5 can be expected in the living brain. Indeed the authors actually show that endogenous mGluR5 is very unlikely to undergo “synaptic recruitment” and enter the PSD, so it is unclear to me what the rapalog experiment tells us about the putative physiological role of synapse-related mGluR5 receptor nano-domains in shaping synaptic signaling.

To be clear, with these experiments we did not intend to mimic a physiological scenario. Although such a scenario cannot be excluded *a priori*, to the best of our knowledge there is no evidence that certain stimuli or signals promote synaptic entry of mGluR5. But, given that there is seemingly a strong biological drive to exclude mGluR5 from the PSD, our main goal was to assess why preventing mGluR5 from entering the PSD is important for synaptic functioning. We therefore reasoned that to test the functional importance of the particular perisynaptic distribution of mGluR5 we should overcome these exclusion mechanisms and force the relocalization of mGluR5

to the synapse. We found that this forced recruitment of mGluR5 to the PSD dramatically increased synaptic calcium transients which, to us, implied the clear functional relevance of the retention of mGluR5 at the perisynaptic zone. These results indicate that the prevention of synaptic entry under physiological conditions is critical for restricting mGluR5 overactivation. We made adjustments to the Results section to better explain this rationale (see line 292-301 in the revised manuscript).

To address this, I suggest the authors seek to corroborate their data by showing that endogenous mGluR5 receptors can have an impact on synaptic calcium signaling (or electrophysiologically measured synaptic signals).

We appreciate this suggestion as we agree that showing that endogenous mGluR5 contributes to mSCTs measured at individual synapses using GCaMP6f is fundamental to corroborate our data. It is well-established that activation of postsynaptic mGluRs modulates spine calcium levels (see references lines 293-294 in revised manuscript). However, to explicitly show that this leads to changes in mSCT frequency we stimulated neurons transfected with GCaMP6f with the group I mGluR-specific agonist DHPG for 5 minutes to activate mGluR5 (with 3 μ M TTX and 0 mM Mg^{2+}). Indeed, we found that the activation of mGluR5 caused a significant, almost 4-fold increase in mSCT frequency (see Figure S8d-f and lines 305-312 in the revised manuscript). Given that dynamic changes in postsynaptic calcium concentration are critical in determining the direction and changes in synaptic strength, this shows that endogenous mGluR5 has a physiological role in shaping synaptic signaling, as has been shown by us before (Scheefhals et al., 2019) and others, reviewed in (Bodzęta et al., 2021; Gerber et al., 2007). Interestingly, recruiting mGluR5 to the PSD also resulted in a 3 to 4-fold increase in mSCT frequency during spontaneous release events. This suggests that the altered calcium signaling is a direct effect of mGluR5 activation.

We agree that the forced re-localization of mGluR5 to the PSD does not represent a physiological scenario, but we hope that the explanation above clarified that we do not claim that this is the case and rather used the heterodimerization tool to study the importance of the synaptic exclusion of mGluR5, also further emphasizing the significance of the C-terminal domain of mGluR5 preventing synaptic entry.

Also, it would be relevant to repeat the current calcium imaging and rapalog experiment while blocking mGluR5 to test whether the altered calcium signaling is indeed a direct effect of mGluR5 and not an indirect effect i.e. from crowding in the PSD.

The suggested explanation could be of interest, perhaps an increase in crowding could potentiate calcium signaling. However, we cannot envision clearly how this would potentiate the activity of the NMDA receptor as we observe in our experiments and feel that exploring this direction is beyond the scope of the present study. Our additional experiments now do demonstrate that mGluR5 has direct effects on calcium signaling in spines (Figure S8d-f).

Other comments

- Relating to my main comment, in Fig 7 only overexpressed mGluR5 are illustrated in the cartoon graphic, is there not still also endogenous mGluR5? The ratio of endogenous to overexpressed mGluR5 receptors is critical for interpreting these experiments and should be somehow quantified, even if by simple immunocytochemistry.

As suggested by the Reviewer, we quantified the ratio of endogenous to overexpressed mGluR5. We transfected neurons with SEP-mGluR5 and stained the total pool (surface and intracellular) of mGluR5 with an mGluR5 antibody. Within the same coverslip we quantified the mGluR5 expression levels in transfected and untransfected neurons and found that mGluR5 overexpression resulted in ~2.1 higher total mGluR5 levels compared to untransfected neurons. mGluR5 contains an ER retention signal preventing the translocation from the ER to the plasma membrane (Roche et al., 1999), a process that is also known to be tightly regulated for overexpressed AMPARs (Kessels et al., 2009; Shi et al., 2001), possibly lowering the overexpression of mGluR5 at the surface further minimizing overexpression artifacts.

- The uPAINT experiments were also performed after overexpression of (SEP-) mGluR5, again without knowing the overexpression level. Could the uPAINT experiments not be performed targeting endogenous mGluR5 using a fluorophore-tagged mGluR5 antibody, or is this not possible?

We agree that tracking endogenous mGluR5 would have been a valuable addition to the manuscript. We attempted to track endogenous mGluR5 tagged with GFP (using CRISPR/Cas9-mediated knock-in, see Figure 1i-k), by labeling GFP with the same anti-GFP nanobody coupled to Atto647N as used in the uPAINT experiments in the manuscript. However, the signal was too poor to reliably track endogenous mGluR5. Therefore, in our experiments we selected neurons with moderate overexpression levels of mGluR5 and confirmed that the localization of endogenous mGluR5 is similar to overexpressed mGluR5 (see Figures 1e-h and S1g-j). As suggested by the Reviewer we also quantified the ratio of endogenous to overexpressed mGluR5, which showed moderate overexpression with a median of 2.1 times higher mGluR5 expression in transfected neurons compared to untransfected neurons (see Figure S1e, f).

- Given the uncertainty of the source of the observed calcium transients, it would be informative to at least confirm they were synaptic NMDAR dependent by investigating if they are blocked by NMDAR blockers/antagonists (or alternatively by including a control group with physiological concentration of Mg in the buffer).

The mSCT analysis as we performed here, has been extensively characterized and validated by several labs and is generally used to record spontaneous activation of NMDA receptors (Metzbower et al., 2019; Reese and Kavalali, 2015). We have independently validated this approach in the lab and confirmed that under basal

conditions mSCTs are principally mediated by NMDA receptors: increasing extracellular calcium concentration or application of the NMDA receptor co-agonist glycine potentiated mSCT frequency, while removing extracellular calcium or application of the NMDAR antagonist AP5 completely blocked mSCTs (data not shown). Here, to test explicitly that the increase in mSCTs by endogenous mGluR5 activation is mediated by NMDA receptors, we applied AP5 following the 5-minute DHPG application (see also description of this experiment above). We found that the robust increase in mSCT frequency following DHPG, was nearly completely eliminated by AP5 (see Figure S8E-G in the revised manuscript). Thus, activation of mGluR5 results in more frequent calcium transients that are mediated by NMDA receptors.

- Line 245: “Although not significant, synaptic mGluR5 Δ C-STGtail diffused at a slower rate compared to mGluR5WT, suggesting that the addition of the STGtail increased the retention in the PSD...”. If the difference is not significant then the authors cannot say that it diffused slower and should modify the text accordingly to be more true to the observed result.

We agree and removed this statement from the text.

Throughout, I suggest showing SD in error bars as it holds valuable information about variation (physiological and methodological), rather than the less informative SEM.

We chose to present the SEM as it is the more generally accepted form to present the variation. We will include a table with all the raw data and calculated SD values for publication.

- In Fig. 1C, perhaps specify that the “ratio” is the “intensity ratio”?

In the legend of Figure 1c, and all other figures where we plot the intensity ratio, we specified that we quantified the ratio of mGluR5 intensity in spine over dendrite.

- Any possible non-normal data should be depicted with median instead of mean, and not with SEM. For example, are data depicted in Fig. S8D normal?

We completely agree with the reviewer and depicted all the non-normal data with median instead of mean.

- Line 77, 78, 79 and throughout: Instead of using simply “anti-PSD-95”, “anti-mGluR5”, and so forth, please spell out whether this refers to an antibody or some other label.

We adjusted the text accordingly.

- Line 303: “form” should be “from”.

We adjusted the text accordingly.

- Line 666: “...room RT,...”

We adjusted the text accordingly.

- Line 807 and throughout text figures and suppl figures: To refer to line profile intensities in 2D images "line profile" or similar is better than the currently used "linescan" or "line scan", which are more often used to refer to dynamic scanning of a line for image acquisition.

We adjusted the text accordingly.

Reviewer #3

Scheefhals et al. aims to determine the behavior of the mGluR5 at the synapse by using super-resolution techniques. The main conclusions are that mGluR5 have some transient trapping at the perisynapse but is mainly excluded from the PSD, the Cterm tail of mGluR5 is implicated in the trapping, the overaccumulation of mGluR5 at the PSD would increase the frequency of calcium entry at spine. The paper uses a wide range of techniques and way of labeling the receptor. The images and the figures are of a high quality. The overall story is very elegant and the conclusion of such studies is strongly awaited by the neuroscience community. Indeed, the dynamic nanoscale organization of both AMPAR and NMDAR has been achieved, but that of mGluR is still lacking. However, some significant experiments/ analyses should be performed to validate the hypothesis.

We thank the reviewer for the considerate assessment and positive evaluation of the manuscript. We have tried to address all the Reviewer's comments in the revised manuscript.

Major points:

Differences between clusters and single particles. With super-resolution, couple of detections close each other can be either a cluster of proteins (as for AMPAR or NMDAR) or the multiple detections due to the blink effect of the molecule or the various frames of a single and immobile track.

We understand from this comment that the Reviewer is concerned that the recurrent blinking of single emitters in SMLM experiments could give rise to the false detection of clusters. We acknowledge this concern and have made sure to prevent misinterpretation of the data in several ways and confirmed in independent experiments the validity of this conclusion. First, as described in the Methods section, as part of the STORM image analysis we track single molecules in consecutive frames to filter out blinking events that remain longer than one frame to prevent overcounting. Second, for the data presented in Figure 2, we defined mGluR5 nanodomains using the DBScan algorithm with stringent criteria, a density cutoff of >480 molecules per μm^2 , epsilon set to 0.35 and >20 localizations, to ensure that nanodomains consisted of a considerable number of receptors, higher than expected from spurious blinking events. To further validate this, we used an alternative, quantitative approach (SR-

Tesseler) to confirm the proper identification of mGluR5 nanodomains in spines (see Figure S2e, f and lines 104-116) in the revised manuscript). Third, the location and characteristics of these nanodomains is not random. We show that these nanodomains are preferentially enriched in the perisynaptic area and now included analysis that nanodomains in spines are significantly larger than in dendrites (lines 114-116), which would not be expected if these arise from random blinking events. Fourth, both the STED imaging and the confinement analysis on the single-molecule tracking data provide experimentally independent assessments of the organization of receptors and both convincingly indicate that receptors are organized in perisynaptic nanodomains. Fifth, we present mechanistic data that manipulation of the CTD of mGluR5 modulates the nanodomain organization of the receptor, further lending support to the observation that the nanodomain organization of mGluR5 we observe is a biological phenomenon and not a misinterpretation or technical artefact.

Moreover, it is now regularly accepted that mGluR are mainly dimeric. It is crucial here to demonstrate whether the objects correspond to a cluster of detections or to a cluster of proteins. The ultimate question is to determine whether there is some transient trapping of a dimer or some well define nano-organization at the perisynapse. This point is crucial for concluding between a clustered organization or a homogeneous organization as argued in Goncalves et al. 2020 in PNAS.

Proper receptor dimerization is a requirement for mGluR5 to pass the quality control system in the ER and to transit along the secretory pathway to form stable, disulphide-linked dimers on the plasma membrane. Therefore, we indeed assume that the majority of labeled receptors resemble functional dimeric receptors and agree that recognizing this is important for the correct interpretation of our data.

Goncalves *et al.*, 2020 present an elegant approach to determine the nanoscale organization of receptors by identifying nano-objects to discriminate between single, dimeric and clustered receptors. We also used SR-Tesseler without set cluster criteria to detect nano-objects to discriminate between dimeric and clustered receptors in the dendritic shaft and spines. In both the dendritic shafts and spines we found many objects, otherwise excluded from the analysis, with the smallest objects likely representing mGluR dimers. Importantly, we found that the nanodomains we identified with DBScan are also detected using SR-Tesseler and that these nanodomains are significantly larger than the nano-objects we found when we set no cluster criteria and that likely outline multiple detections arising from single emitters (see lines 104-116 in Results section of revised manuscript). As also outlined above, based on the collective data from several independent experimental approaches and new analysis using independent approaches in both fixed and live neurons, we conclude that the distribution of mGluR5 is not homogeneous, but that mGluR5 is transiently confined in perisynaptic nanodomains.

Around the same idea, the distribution of the size of the individual mGluR5 is not described, the pointing accuracy of a single detection and the resolution of a single particle differs. In the figures, in STED and STORM, many 80 nm clusters resemble to

single molecules. It might be worth comparing the synaptic “cluster/ single molecule” with that on dendrite.

We included new analysis showing that nanodomains in spines are significantly larger than in dendrites (lines 114-116).

The ring analysis (figure 2E) is a bit confusing to me, if the PSD is on 2 rings (around 200 nm), it is surrounded by the two perisynaptic rings (200 nm from each side), it makes 600 nm which is the size of the spine. Thus, the other rings tend to exhibit a lower density of mGluR because it is out of the neuron (except at the neck localization).

This is a good point and is indeed a disadvantage of the ring analysis. However, the main conclusion here is that mGluR5 is largely absent from the PSD and is enriched surrounding the PSD. We agree that the extrasynaptic rings in some instances extend outside the spine and tend to exhibit a lower density. However, Figure S2b also shows that the outer rings are still largely overlapping with the spine, and not just at the neck. Moreover, Figure S2c which shows the same data as Figure 2e but plots the absolute number of mGluR5 localizations (not corrected for the area of the rings), shows the number of localizations detected in the outer extrasynaptic rings. In addition, we show the absolute distance of localization and clusters to the border of the PSD in Figure 2d and h. Together with the observation that mGluR5 is organized in subsynaptic nanodomains, this shows that mGluR5 is not just simply excluded from the PSD but has a high degree of organization in perisynaptic nanodomains that is regulated by the CTD. Importantly, Figures 3 and 4 corroborate these observations in live neurons by showing that mGluR5 single-molecule tracks are transiently confined in perisynaptic nanodomains.

- The vast majority of the experiments are performed by overexpressing the mGluR5 which can affect the density and distribution, but more importantly, by putting a GFP (even in the elegant knock in experiments) which have a quite large size. If we had the size of the GFP to the size of the extracellular part of the mGluR, is it possible that a form of exclusion is created at the PSD level which is already full of adhesion proteins and where the presynapse is probably the closest from the postsynapse?

We understand the concern of the Reviewer, however we do not think that the (mild) overexpression of mGluR5 in these experiments or the size imposed by the GFP tag influence the exclusion of mGluR5 from the PSD we observe. First, we studied the localization of mGluR5 and its exclusion from the PSD with several approaches: overexpression of the receptor with different tags (SEP, Halo), different imaging modalities (live-cell tracking and fixed STORM and STED), using CRISPR/Cas9-mediated tagging of endogenous mGluR5 and by immunostaining for endogenous mGluR5 (Figure S1e-h) – all confirming that wild-type mGluR5 is excluded from the PSD, consistent with the literature. Second, we do not find that the addition of the GFP (or SEP) tag to a receptor hinders synaptic entry. Notably, in Figure S1l-o we show that recombinant SEP-GluA2 does enter the PSD and co-localizes with Homer1c, as expected. Of note, the extracellular domain of AMPARs is considerably bulkier than that of mGluR5, as it contains two globular extracellular structures, forms tetramers

and extends 13 nm into the synaptic cleft, whereas mGluRs consist of one extracellular structure and a small cysteine-rich region, forms dimers and extends 10 nm into the synaptic cleft. Third, we found that the mGluR5DC-STGtail chimeric receptor is strongly enriched at the PSD, while the mGluR5DC mutant is more diffuse and cross the synapse, further indicating that the SEP tag on itself does not prevent entry of mGluR5 to the synapse.

On the same line, if the exclusion of the PSD (which has been already shown in various paper), is combined to the synaptic enrichment due to local endo/ exocytosis, and the shape of the neck which limit the exchange between spine and dendrite, it is possible that local enrichment at the perisynapse is only a passive result and not a specific trapping. This does not change the quality of the experiments but more their interpretations.

This is a very interesting suggestion. We also considered the possibility that mGluR5 is simply being excluded from the PSD due to the molecular crowded PSD, and as the Reviewer also rightfully points out that lateral diffusion of mGluR5 might be restricted by the spatial constraints imposed by spine shape and in particular the thin spine neck. To address this, we designed a few key experiments, which include the single-molecule localization cluster analysis and the single-molecule tracking confinement analysis. Importantly, we observed that mGluR5 was not homogeneously distributed around the PSD, but assembled in distinct perisynaptic nanodomains, suggesting that specific mechanisms hinder mGluR5 diffusion at the perisynaptic zone. We also showed that these mechanisms are mediated by the C-terminal domain (CTD) of mGluR5. However, the perisynaptic nanodomains are indeed not necessarily directly regulated by protein-protein interactions, but other mechanisms such as cytoskeletal hindrance could also mediate the organizational properties of the mGluR5 CTD (as discussed at lines 363-367 in the revised manuscript). More specifically, perisynaptic actin could also act as a gatekeeper, imposing a diffusional barrier to receptor entry into the PSD (Burette et al., 2012; Frost et al., 2010; Morone et al., 2006) and the actin filamentous meshwork may compartmentalize proteins at the perisynaptic membrane (Sungkaworn et al., 2017).

- The u-PAINT experiments are well done and important, the transient trapping of some receptors is clearly demonstrated. However, there is some bias in the analysis. (1) It is normal that the transient perisynaptic move more because they have been chosen on this property, they are not trapped at the perisynapse, so they are more mobile.

Perisynaptic and transient perisynaptic trajectories have in common that they, based on the set criteria, are associated with the perisynaptic zone. However, to show that there are two types of perisynaptic receptor pools that behave dynamically different, we defined these two categories based on the percentage of overlap with the perisynaptic zone. Perisynaptic tracks had to overlap $\geq 60\%$ with the perisynaptic zone and transient perisynaptic tracks overlapped $>0\%$ and $<60\%$ with the perisynaptic zone. The Reviewer rightfully points out that the higher mobility found for the transient perisynaptic trajectories could also be a result of the fact that they are less limited in their diffusion area. Indeed, our hypothesis is that the transient perisynaptic trajectories

are more mobile compared to the perisynaptic trajectories due to specific mechanisms at the perisynaptic zone that hinder mGluR5 diffusion. However, in principle mechanisms present at the extrasynaptic area could also trap the transient perisynaptic trajectories and are therefore not necessarily expected to be more mobile. Since we show that the transient perisynaptic trajectories are considerably more mobile (Figure 3e, f) and the number of confinement zones decrease further away from the perisynaptic zone (Figure 4j), the degree of organization outside the perisynaptic zone is indeed less.

It could be interesting to have the distribution of the dendritic receptors, and more particularly on the knock in ones, to avoid artifact from overexpression. Are there trapped receptors on the dendrites?

This point has also been raised by Reviewer #2. We attempted to track the mGluR5 GFP knock-in, however were unsuccessful in obtaining single-molecule localization of sufficient quality to reliably reconstruct the receptor trajectories. Therefore, we always selected neurons with moderate overexpression levels of mGluR5 and confirmed that the localization of endogenous mGluR5 is similar to overexpressed mGluR5 (see Figures 1e-h and S1e-h). Reviewer #2 suggested to quantify the ratio of endogenous to overexpressed mGluR5, which showed moderate overexpression with a median of 2.10 times higher mGluR5 expression in transfected neurons compared to untransfected neurons (see also our reply to Reviewer #2 and Figure S1e, f).

Even though we believe that characterizing mGluR organization in the dendritic shaft, including at inhibitory synapses, will further our understanding of the tight relation between the nanoscale mGluR receptor positioning and signaling, this is beyond the scope of this manuscript. We do want to point out that a small fraction of immobile and confined mGluRs was localized further away from the border of the PSD towards extrasynaptic regions (see Figure 4j and k) and also expect to find trapped receptors on the dendrites.

(2) The MSD curve of the figure 3H (corresponding to the legend 3I) start at the 0 value which is not normal for a MSD which should start from a positive value corresponding to the pointing accuracy. And the curves seem to be a fit of the data, what is use for this fit?

The value of 0 at MSD(0) was added to prevent negative-slope fits, which is especially apparent for molecules with a low D_{eff} , similar to what has been previously reported by Lu *et al.*, 2014 (Lu et al., 2014). The presented curve is not a fit of the data but shows the average of all MSDs (n = number of neurons) and only the first part of the MSD plot is shown up to $dt = 30$ frames.

Concerning the role of the C-term of mGluR, I wonder whether the correct control might not be the delta-C-term fused to the STG tail, because, of course the fusion of the entire tail of stargazin to the already long tail of mGluR could have some exclusion effects at the PSD because of the size. Another possibility could be to add a PSD binding site at the end of the mGluR tail.

This point has also been raised by Reviewer #1. We followed the Reviewers' suggestion and fused the mGluR5 CTD to the STGtail, with the exception of the STGtail PDZ-binding motif as explained in the revised manuscript, see lines 252-270. We found that the mGluR5 CTD is sufficient to prevent the synaptic entry of the normally strongly enriched STGtail, further confirming the hypothesis that the mGluR5 CTD mediates synaptic exclusion. These data are now included in the revised manuscript in Supplementary Figure 6.

Finally, the effect of the recruitment of mGluR at the PSD. These experiments are difficult and clever. The total amount of mGluR per spine need to be done. For now, it is not known if the increase of calcium is due to the larger amount of mGluR at the spine or to its re-localization at the PSD.

The resolution of these live-cell confocal images is not sufficient to make a strict and clear distinction between the amount of mGluR5 localized at spines or at the PSD. However, when considering the representative images in Figure 7, the dendritic pool of mGluR5 does not seem to change upon the addition of rapalog. This indicates that mGluRs are largely re-localized from (perisynaptic) regions in the spine towards the center of the PSD, causing the increase in calcium transients.

As has been shown by multiple computational models, spontaneous release events produce a very brief and narrow peak of glutamate in the synaptic cleft, which is restricted to a small area (<100 nm). As a results of these biophysical properties, only receptors near the vesicle release site are predicted to become activated and participate in synaptic calcium signaling. Therefore, we assume that the increase in mSCT frequency is due to an increase of mGluRs at the synapse specifically, and not due to a general increase of mGluRs in spines.

Finally, it is difficult to understand why the frequency is affected

We interpret the increase in mSCT frequency as an increase in the probability that a single, spontaneous synaptic release event activates mGluRs and elicits a calcium transient. We cannot provide a definitive explanation for the exact effects downstream of mGluR5 activation, as also discussed in response to other Reviewers, but we do now provide evidence that the effects are mediated almost completely by the potentiation of NMDA receptor activity (Figure S8d-f). Although speculative, one could also predict that calcium release from internal stores downstream of mGluR5 activation amplifies NMDA receptor-mediated calcium transients increasing the number of detectable mSCTs (see also Figure 8h). Indeed a critical role has been shown for calcium-induced calcium release in amplifying NMDA receptor-mediated mSCTs during spontaneous events (Reese and Kavalali, 2015).

REFERENCES

- Bodzęta, A., Scheefhals, N., MacGillavry, H.D., 2021. Membrane trafficking and positioning of mGluRs at presynaptic and postsynaptic sites of excitatory synapses. *Neuropharmacology* 200, 108799.
- Burette, A.C., Lesperance, T., Crum, J., Martone, M., Volkman, N., Ellisman, M.H., Weinberg, R.J., 2012. Electron tomographic analysis of synaptic ultrastructure. *Journal of Comparative Neurology* 520, 2697-2711.
- Frost, N.a., Shroff, H., Kong, H., Betzig, E., Blanpied, T.a., 2010. Single-molecule discrimination of discrete perisynaptic and distributed sites of actin filament assembly within dendritic spines. *Neuron* 67, 86-99.
- Gerber, U., Gee, C.E., Benquet, P., 2007. Metabotropic glutamate receptors: intracellular signaling pathways. *Current Opinion in Pharmacology* 7, 56-61.
- Kessels, H.W., Kopec, C.D., Klein, M.E., Malinow, R., 2009. Roles of stargazin and phosphorylation in the control of AMPA receptor subcellular distribution. *Nat Neurosci* 12, 888-896.
- Lu, H.E., MacGillavry, H.D., Frost, N.A., Blanpied, T.A., 2014. Multiple spatial and kinetic subpopulations of CaMKII in spines and dendrites as resolved by single-molecule tracking PALM. *J Neurosci* 34, 7600-7610.
- Metzbower, S., Joo, Y., Benavides, D., Blanpied, T., 2019. Properties of individual hippocampal synapses influencing NMDA-receptor activation by spontaneous neurotransmission. Properties of individual hippocampal synapses influencing NMDA-receptor activation by spontaneous neurotransmission 6, 590141.
- Morone, N., Fujiwara, T., Murase, K., Kasai, R.S., Ike, H., Yuasa, S., Usukura, J., Kusumi, A., 2006. Three-dimensional reconstruction of the membrane skeleton at the plasma membrane interface by electron tomography. *Journal of Cell Biology* 174, 851-862.
- Reese, A.L., Kavalali, E.T., 2015. Spontaneous neurotransmission signals through store-driven Ca²⁺ transients to maintain synaptic homeostasis. *eLife* 4, 1-15.
- Roche, K.W., Tu, J.C., Petralia, R.S., Xiao, B., Wenthold, R.J., Worley, P.F., 1999. Homer 1b Regulates the Trafficking of Group I Metabotropic Glutamate Receptors * constitutively expressed splice form of the immediate. *Science* 274, 25953-25957.
- Scheefhals, N., Catsburg, L.A.E., Westerveld, M.L., Blanpied, T.A., Hoogenraad, C.C., Macgillavry, H.D., Scheefhals, N., Catsburg, L.A.E., Westerveld, M.L., Blanpied, T.A., Hoogenraad, C.C., 2019. Shank Proteins Couple the Endocytic Zone to the Postsynaptic Density to Control Trafficking and Signaling of Metabotropic Glutamate Receptor 5 Report Shank Proteins Couple the Endocytic Zone to the Postsynaptic Density to Control Trafficking and Signaling. *CellReports* 29, 258-269.e258.
- Shi, S., Hayashi, Y., Esteban, J.A., Malinow, R., 2001. Subunit-specific rules governing AMPA receptor trafficking to synapses in hippocampal pyramidal neurons. *Cell* 105, 331-343.
- Sungkaworn, T., Jobin, M.-L., Burnecki, K., Weron, A., Lohse, M.J., Calebiro, D., 2017. Single-molecule imaging reveals receptor-G protein interactions at cell surface hot spots. *Nature*.

REVIEWER COMMENTS

Reviewer #1 (Remarks to the Author):

The authors responded to referees' questions with appropriate experiments, and they also followed our advices and modified the discussion to avoid over-interpreting their results. This greatly improves the quality of the paper.

Reviewer #2 (Remarks to the Author):

The authors have done a great job revising the manuscript, and have adequately addressed the concerns I had. The newly performed experiments, along with their consideration to the reviewer comments, add substantially to the story, and my reservations about the physiological relevance of forcing mGluR5 into the synapse no longer stand.

Indeed, the manuscript appears solid and balanced in the way it introduces and discusses the new discoveries about mGluR5 location and function in synaptic signaling.

Minor comments (DBScan) is abbreviated twice (lines 89 and 98).

Line 134: In the given context, "Homer1c expression..." should here correctly be "Homer1c overexpression..."

Reviewer #3 (Remarks to the Author):

The authors answered to all my requests. I consider the paper suitable for publication at Nature communication

REVIEWERS' COMMENTS

Reviewer #1 (Remarks to the Author):

The authors responded to referees' questions with appropriate experiments, and they also followed our advices and modified the discussion to avoid over-interpreting their results. This greatly improves the quality of the paper.

We thank the reviewer for the careful assessment and positive evaluation of the revised manuscript.

Reviewer #2 (Remarks to the Author):

The authors have done a great job revising the manuscript, and have adequately addressed the concerns I had. The newly performed experiments, along with their consideration to the reviewer comments, add substantially to the story, and my reservations about the physiological relevance of forcing mGluR5 into the synapse no longer stand.

Indeed, the manuscript appears solid and balanced in the way it introduces and discusses the new discoveries about mGluR5 location and function in synaptic signaling.

We thank the reviewer for the careful assessment and positive evaluation of the revised manuscript.

Minor comments (DBScan) is abbreviated twice (lines 89 and 98).

Adjusted accordingly.

Line 134: In the given context, "Homer1c expression..." should here correctly be "Homer1c overexpression...".

Adjusted accordingly.

Reviewer #3 (Remarks to the Author):

The authors answered to all my requests. I consider the paper suitable for publication at Nature communication

We thank the reviewer for the careful assessment and positive evaluation of the revised manuscript.